# Sub-linear Regret Bounds for Bayesian Optimisation in Unknown Search Spaces

**Hung Tran-The,**[*] **Sunil Gupta, Santu Rana, Huong Ha, Svetha Venkatesh**
Applied Artificial Intelligence Institute
Deakin University, Australia

## Abstract

Bayesian optimisation is a popular method for efficient optimisation of expensive black-box functions. Traditionally, BO assumes that the search space is known. However, in many problems, this assumption does not hold. To this end, we propose a novel BO algorithm which expands (and shifts) the search space over iterations based on controlling the expansion rate thought a *hyperharmonic series*. Further, we propose another variant of our algorithm that scales to high dimensions. We show theoretically that for both our algorithms, the cumulative regret grows at sub-linear rates. Our experiments with synthetic and real-world optimisation tasks demonstrate the superiority of our algorithms over the current state-of-the-art methods for Bayesian optimisation in unknown search space.

## 1 Introduction

Bayesian optimisation (BO) is a powerful and flexible tool for efficient global optimisation of expensive black-box functions. An underlying limitation of existing approaches is that the search is restricted to a pre-defined and fixed search space, thus implicitly assuming that this search space will contain the global optimum. To set suitable bounds of the search space, prior knowledge is required. When exploring entirely new problems (e.g a new machine learning model), such prior knowledge is often poor, and thus the specification of the search space can be erroneous leading to suboptimal solutions. For example, in many machine learning algorithms we have hyperparameters or parameters that can take values in an unbounded space e.g. $L1/L2$ penalty hyperparameters in elastic-net can take any nonnegative value. Similarly, the weights of a neural network can take any real values. No matter how large a finite search space is set, one cannot be sure if this search space contains the global optimum.

This problem is considered in [23, 17, 18, 8, 2], and UBO [8] is the first to provide global convergence analysis. However, they consider a weak version of the global convergence, i.e., instead of seeking the exact global optimum, they find a solution wherein the function value is within $\epsilon > 0$ of the global optimum. Further, there is no analysis on the convergence rate of this algorithm which is important for understanding the efficiency of the optimisation.

Another complication arises when high dimensional problems are considered (e.g. hyperparameter tuning [24], reinforcement learning [1]), as BO scales poorly in practice. With unknown search spaces, we need to consider evolving/growing search spaces. This growth in search spaces makes high dimensional BO further challenging as the search space is already exponentially large with respect to dimensions. Both these challenges compound the difficulty in maximising the acquisition functions. With limited budgets, the accuracy of points suggested by the acquisition step is often poor and this adversely affects both the convergence and the efficiency of the BO algorithm. Thus solutions to BO in unknown high-dimensional search spaces need to be found.

---

[*]Correspondence to: Hung Tran-The <`hung.tranthe@deakin.edu.au`>.

In this paper, we address these open problems. Our contributions are as follows:

- We introduce a novel BO algorithm for unknown search space, using a volume expansion strategy with a rate of expansion controlled through *hyperharmornic series* [3]. We show that our algorithm achieves a sub-linear convergence rate.

- We then provide a first solution for BO problem with unknown high dimensional search spaces. Our solution is based on using a restricted search space consisting of a set of hypercubes with small sizes. Based on controlling the number of hypercubes according to the expansion rate of the search space, we derive an upper bound on the cumulative regret and theoretically show that it can achieve a sub-linear growth rate.

- We evaluate our algorithms extensively using a variety of optimisation tasks including optimisation of several benchmark functions and tuning both the hyperparameters (Elastic Net) and parameters of machine learning algorithms (weights of a neural network and Lunar Lander). We demonstrate that our algorithms have better sample and computational efficiency compared to existing methods on both synthetic and real optimisation tasks. Our source code is publicly available at `https://github.com/Tran-TheHung/Unbounded_Bayesian_Optimisation`.

## 2 Related Work

There are two main approaches in previous work addressing BO with unknown search spaces. The first tackles the problem by nullifying the need to declare the search space - instead a regularized acquisition function is optimised on an unbounded search space such that its maximum can never be at infinity. However, this approach requires critical parameters that are difficult to specify in practice, and there is no theoretical guarantee on the optimisation efficiency. The second approach uses volume expansion - starting from a user-defined region, the search space is sequentially expanded during optimisation. The simplest strategy repeatedly doubles the volume of the search space every few iterations [23]. Such a strategy is not efficient as it grows the search space exponentially. [18] propose a expansion strategy based on a filter, however they require an additional crucial assumption that the initial search space is sufficiently close to the optimum. Further, their regret bound is non vanishing. More recently, [2] propose an adaptive expansion strategy based on the uncertainty of the GP model, but do not provide convergence guarantees. A recent approach by [8] is the first to provide a global convergence analysis. However, their work has two limitations. First, the convergence analysis aims at $\epsilon$-regret, meaning the algorithm only converges approximately. Second, there is no analysis of the convergence rate. Compared to these works, our approach is novel and is only one to guarantee the sub-linear convergence rate.

In another context, the high dimensional BO has been studied extensively in the literature. In order to make BO scalable to high dimensions, most of the methods make restrictive structural assumptions such as the function having an effective low-dimensional subspace [27, 4, 7, 5, 28, 16], or being decomposable in subsets of dimensions [11, 13, 22, 15, 10]. Through these assumptions, the acquisition function becomes easier to optimise and the global optimum can be found. However, such assumptions are rather strong. Without these assumptions, high-dimensional BO problem is more challenging. There have been a limited attempts to develop scalable BO methods [19, 12, 6, 26]. To our knowledge, all these works have not been considered in the case of unknown search spaces. We provide the first solution for the unknown high-dimensional problem. Our solution does not make structural assumptions on the function.

## 3 Preliminaries

Bayesian optimisation (BO) finds the global optimum of an unknown, expensive, possibly non-convex function $f(x)$. It is assumed that we can interact with $f$ only by querying at some $\mathbf{x} \in \mathbb{R}^d$ and obtain a noisy observation $y = f(\mathbf{x}) + \epsilon$ where $\epsilon \sim \mathcal{N}(0, \sigma^2)$. The search space is required to specified a priori and is assumed to include the true global optimum. BO proceeds sequentially in an iterative fashion. At each iteration, a surrogate model is used to probabilistically model $f(\mathbf{x})$. Gaussian process (GP) [21] is a popular choice for the surrogate model as it offers a prior over a large class of functions and its posterior and predictive distributions are tractable. Formally, we have $f(\mathbf{x}) \sim \mathcal{GP}(m(\mathbf{x}), k(\mathbf{x}, \mathbf{x}'))$ where $m(\mathbf{x})$ and $k(\mathbf{x}, \mathbf{x}')$ are the

mean and the covariance (or kernel) functions. Popular covariance functions include Squared Exponential (SE) kernels, Matérn kernels etc. Given a set of observations $\mathcal{D}_{1:t} = \{\mathbf{x}_i, y_i\}_{i=1}^t$, the predictive distribution can be derived as $P(f_{t+1}|\mathcal{D}_{1:t}, \mathbf{x}) = \mathcal{N}(\mu_{t+1}(\mathbf{x}), \sigma_{t+1}^2(\mathbf{x}))$, where $\mu_{t+1}(\mathbf{x}) = \mathbf{k}^T[\mathbf{K} + \sigma^2\mathbf{I}]^{-1}\mathbf{y} + m(\mathbf{x})$ and $\sigma_{t+1}^2(\mathbf{x}) = k(\mathbf{x}, \mathbf{x}) - \mathbf{k}^T[\mathbf{K} + \sigma^2\mathbf{I}]^{-1}\mathbf{k}$. In the above expression we define $\mathbf{k} = [k(\mathbf{x}, \mathbf{x}_1), ..., k(\mathbf{x}, \mathbf{x}_t)]$, $\mathbf{K} = [k(\mathbf{x}_i, \mathbf{x}_j)]_{1 \le i, j \le t}$ and $\mathbf{y} = [y_1, \ldots, y_t]$.

After the modeling step, an acquisition function is used to suggest the next $\mathbf{x}_{t+1}$ where the function should be evaluated. The acquisition step uses the predictive mean and the predictive variance from the surrogate model to balance the exploration of the search space and exploitation of current promising regions. Some examples of acquisition functions include Expected Improvement (EI) [14], GP-UCB [25] and PES [9]. We use GP-UCB acquisition function which is defined as

$$u_t(\mathbf{x}) = \mu_{t-1}(\mathbf{x}) + \sqrt{\beta_t}\sigma_{t-1}(\mathbf{x}), \tag{1}$$

where $\beta_t$ balances the exploration and the exploitation (see [25]).

**Cumulative Regret:** To measure the performance of a BO algorithm, we use the regret, which is the loss incurred by evaluating the function at $x_t$, instead of at unknown optimal input, formally $r_t = f(\mathbf{x}^*) - f(\mathbf{x}_t)$. The cumulative regret is defined as $R_T = \sum_{1 \le t \le T} r_t$ , the sum of regrets incurred over given a horizon of $T$ iterations. If we can show that $\lim_{T \to \infty} \frac{R_T}{T} = 0$, the cumulative regret is **sub-linear**, and so the algorithm efficiently converges to the optimum.

# 4    Problem Setup and HuBO Algorithm

Bayesian optimisation aims to find the global optimum of black-box functions, i.e.

$$\mathbf{x}^* = \text{argmax}_{\mathbf{x} \in \mathbb{R}^d} f(\mathbf{x}),$$

where $d$ is the input dimension of the search space. Differing from traditional BO where the search space is assumed to be known a priori, we assume that the search space is **unknown**. As in [8], we assume that $\mathbf{x}^*$ is not at infinity to make the BO tractable.

When the search space is unknown, one heuristic solution is to specify it arbitrarily. However, there are two problems: (1) an arbitrary search space that is finite, no matter how large, may not contain the global optimum (2) optimisation efficiency decreases with increasing size of the search space.

We propose a volume expansion strategy such that the search space can eventually cover the whole $\mathbb{R}^d$ (therefore, guaranteed to contain unknown $\mathbf{x}^*$), while the expansion rate is kept slow enough so that the algorithm efficiently converges, i.e. $\lim_{T \to \infty} \frac{\sum_{1 \le t \le T}(\bar{f}(\mathbf{x}^*) - f(\mathbf{x}_t))}{T} \to 0$, given any $T > 0$. To do this, our key idea is to iteratively expand and shift the search space toward the "promising regions". At iteration $t$, we expand the search space by $\mathcal{O}(t^\alpha)$, where $\alpha < 0$. We choose this form so that the search space expansion slows over time. The parameter $\alpha$ is set to guarantee the efficient convergence. Our volume expansion strategy is as follows: starting from an initial user-defined region, denoted by $\mathcal{X}_0 = [a, b]^d$, the search space at iteration $t$, denoted by $\mathcal{X}_t = [a_t, b_t]^d$ will be built from $\mathcal{X}_{t-1} = [a_{t-1}, b_{t-1}]^d$ by a sequence of transformations as follows:

$$\mathcal{X}_{t-1} \to \mathcal{X}_t' \to \mathcal{X}_t \tag{2}$$

where $\mathcal{X}_t' = [a_t', b_t']^d$ is expanded from $\mathcal{X}_{t-1}$ by the $(\frac{b-a}{2})t^\alpha$ increment in each direction for all dimensions as $a_t' = a_{t-1} - \frac{b-a}{2}t^\alpha$ and $b_t' = b_{t-1} + \frac{b-a}{2}t^\alpha$.

To build $\mathcal{X}_t$ from $\mathcal{X}_t'$, we translate the center of $\mathcal{X}_t'$, denoted by $\mathbf{c}_t'$ toward the best solution found until iteration $t$. To avoid a "fast" translation of $\mathbf{c}_t'$, which could cause the divergence, we use a fixed, finite domain $\mathcal{C}_{initial}$ to restrict the translation of $\mathbf{c}_t'$. We translate $\mathbf{c}_t'$ toward a point $\mathbf{c}_t$ where $\mathbf{c}_t \in \mathcal{C}_{initial}$ is the closest point to the best solution found until iteration $t$. In practice, our algorithm would typically benefit by setting a large $\mathcal{C}_{initial}$ as this allows the search space in iteration $t$ to be centred close to the best found solution. However, irrespective of the size of $\mathcal{C}_{initial}$, as we show in our convergence analysis, our search space expansion scheme is still guaranteed to converge to $\mathbf{x}^*$.

In this transformation, step $\mathcal{X}_{t-1} \to \mathcal{X}_t'$ plays the role to expand the search space. Step $X_t' \to \mathcal{X}_t$ plays the role to translate the search space towards the promising region surrounding the best solution found so far. By induction, we can compute the volume of $\mathcal{X}_t$ as $Vol(\mathcal{X}_t)$: $Vol(\mathcal{X}_t) =$

---
**Algorithm 1** HuBO Algorithm
---
**Parameters**: $\alpha \in \mathbb{R}$ - rate of expanding the volume of the search space
**Initialisation**: Define an initial search space $\mathcal{X}_0 = [a, b]^d$, a finite domain $\mathcal{C}_{initial} = [c_{min}, c_{max}]^d$, where $\mathcal{X}_0 \subseteq \mathcal{C}_{initial}$. Sample initial points in $\mathcal{X}_0$ to build $\mathcal{D}_0$.

1: **for** $t = 1, 2, ...T$ **do**
2:      Fit the Gaussian process using $\mathcal{D}_{t-1}$.
3:      Define $\mathcal{X}_t$ using (2).
4:      Find $\mathbf{x}_t = \text{argmax}_{\mathbf{x} \in \mathcal{X}_t} u_t(\mathbf{x})$, where $u_t(\mathbf{x})$ defined as in Eq (1) to find $\mathbf{x}_t$.
5:      Sample $y_t = f(\mathbf{x}_t) + \epsilon_t$.
6:      Augment the data $\mathcal{D}_t = \{\mathcal{D}_{t-1}, (\mathbf{x}_t, y_t)\}$.
7: **end for**
---

$(b - a)^d(1 + \sum_{j=1}^{t} j^\alpha)^d$. Therefore, given any $t$, the volume of the search space $\mathcal{X}_t$ is controlled by a partial sum of a *hyperharmonic series* $\sum_{j=1}^{t} j^\alpha$ [3].

Our strategy called the **H**yperharmonic **u**nbounded **B**ayesian **O**ptimisation(HuBO) is described in Algorithm 1. It closely follows the standard BO algorithm. The only difference lies in the acquisition step where instead of using a fixed search space, the search space expands in each iteration following (2). We use the GP-UCB acquisition function with $\beta_t = 2log(4\pi_t/\delta) + 4dlog(dts_2(b-a)(1 + \sum_{j=1}^{t} j^\alpha)\sqrt{log(4ds_1/\delta)})$, where $\sum_{t \geq 1} \pi_t^{-1} = 1, \pi_t > 0$.

### 4.1 Convergence Analysis of HuBO Algorithm

In this section, we provide the convergence analysis of proposed HuBO Algorithm. **All proofs are provided in the Supplementary Material**. To guarantee the convergence, the first necessary condition is that the search space eventually contains $\mathbf{x}^*$.

**Theorem 1** (Reachability). If $\alpha \geq -1$, then the HuBO algorithm guarantees that there exists a constant $T_0 > 0$ (independent of $t$) such that when $t > T_0$, $\mathcal{X}_t$ contains $\mathbf{x}^*$.

*Proof.* We denote the center of the user-defined finite region $\mathcal{C}_{initial} = [c_{min}, c_{max}]^d$ as $\mathbf{c}_0$. By the assumption of $\mathbf{x}^*$ being not at infinity, there exists a smallest range $[a_g, b_g]^d$ so that both $\mathbf{x}^*$ and $\mathbf{c}_0$ belong to $[a_g, b_g]^d$. By induction, the search space $\mathcal{X}_t$ at iteration $t$ is a hypercube, denoted by $[a_t, b_t]^d$. Following our search space expansion, the center of $\mathcal{X}_t$ only moves in region $\mathcal{C}_{initial}$. Therefore, for each dimension $i$, we have in the worst case, $(b_t - [\mathbf{c}_0]_i)$ is at least $\frac{c_{min} - c_{max}}{2} + \frac{b_t - a_t}{2}$ and $([\mathbf{c}_0]_i - a_t)$ is at most $\frac{c_{max} - c_{min}}{2} - \frac{b_t - a_t}{2}$. By induction, we can compute the length of $\mathcal{X}_t$ as $b_t - a_t$ : $b_t - a_t = (b - a)(1 + \sum_{j=1}^{t} j^\alpha)$. Therefore, $(b_t - [\mathbf{c}_0]_i)$ is at least $\frac{c_{min} - c_{max}}{2} + \frac{b-a}{2}(1 + \sum_{j=1}^{t} j^\alpha)$ and $([\mathbf{c}_0]_i - a_t)$ is at most $\frac{c_{max} - c_{min}}{2} - \frac{b-a}{2}(1 + \sum_{j=1}^{t} j^\alpha)$.

If there exists a $T_0$ such that two conditions satisfy: (1) $\frac{c_{min} - c_{max}}{2} + \frac{b-a}{2}(1 + \sum_{j=1}^{T_0} j^\alpha) \geq b_g$, and (2) $\frac{c_{max} - c_{min}}{2} - \frac{b-a}{2}(1 + \sum_{j=1}^{T_0} j^\alpha) < a_g$, then we can guarantee that for all $t > T_0$, the search space $\mathcal{X}_t$ will contain $[a_g, b_g]^d$ and thus also contain $\mathbf{x}^*$. Such a $T_0$ exists because $\sum_{j=1}^{t} j^\alpha$ is a diverging sum with $t$ when $\alpha \geq -1$ [3]. From the conditions (1) and (2), we can see that $T_0$ is a function of parameters $a, b, c_{min}, c_{max}, \alpha$ and $a_g, b_g$. We provide the complete proof in the Supplementary. $\square$

Using the existence of $T_0$, we derive a cumulative regret for our proposed HuBO algorithm by applying the techniques of GP-UCB as in [25], however, with an adaptation according to the growth of search spaces over time.

**Theorem 2** (Cumulative Regret $R_T$ of HuBO Algorithm). Let $f \sim \mathcal{GP}(\mathbf{0}, k)$ with a stationary covariance function $k$. Assume that there exist constants $s_1, s_2 > 0$ such that $\mathbb{P}[sup_{\mathbf{x} \in \mathcal{X}} |\partial f / \partial x_i| > L] \leq s_1 e^{-(L/s_2)^2}$ for all $L > 0$ and for all $i \in \{1, 2, ..., d\}$. Pick a $\delta \in (0, 1)$. Thus, if $-1 \leq \alpha < 0$ then for any horizon $T > T_0$, the cumulative regret of the proposed HuBO algorithm is bounded as

- $R_T \leq \mathcal{O}^*(T^{\frac{(\alpha+1)d+1}{2}})$ if $k$ is a SE kernel,

- $R_T \leq \mathcal{O}^*(T^{\frac{d^2(\alpha+2)+d}{4\nu+2d(d+1)}})$ if $k$ is a Matérn kernel

---
**Algorithm 2** HD-HuBO Algorithm
---
**Parameters**: $\alpha \in \mathbb{R}$- rate of expanding the search space, $\lambda \in \mathbb{R}^+$ and $N_0$- the parameters related to the number of hypercubes, $l_h \in \mathbb{R}^+$- the size of hypercubes

**Initialisation**: Define an initial space $\mathcal{X}_0 = [a, b]^d$, an initial domain $\mathcal{C}_{initial} = [c_{min}, c_{max}]^d$, where $\mathcal{X}_0 \subseteq \mathcal{C}_{initial}$. Sample initial points in $\mathcal{X}_0$ to construct $\mathcal{D}_0$.

1: **for** $t = 1, 2, ...T$ **do**
2:      Fit a Gaussian process using $\mathcal{D}_{t-1}$.
3:      Update the search space $\mathcal{H}_t = \{H(\mathbf{z}_t^1, l_h) \cup ... \cup H(\mathbf{z}_t^{N_t}, l_h)\} \cap \mathcal{X}_t$, where $N_t = N_0\lceil t^\lambda \rceil$ and
       $N_t$ values of $\mathbf{z}_t^i$ are drawn uniformly at random from $\mathcal{X}_t$.
4:      Find $\mathbf{x}_t = \text{argmax}_{\mathbf{x} \in \mathcal{H}_t} u_t(\mathbf{x})$
5:      Sample $y_t = f(\mathbf{x}_t) + \epsilon_t$.
6:      Augment the data $\mathcal{D}_t = \{\mathcal{D}_{t-1}, (\mathbf{x}_t, y_t)\}$
7: **end for**
---

with probability greater than $1 - \delta$.

**Sub-linear Regret**    By Theorem 2, the HuBO algorithm obtains a sub-linear cumulative regret for SE kernels if $-1 \leq \alpha < -1 + \frac{1}{d}$, and for Matérn kernels if $-1 \leq \alpha < \min\{0, -1 + \frac{2\nu}{d^2}\}$.

## 5   HD-HuBO Algorithm in High Dimensions

Further, we extend the HuBO algorithm for high dimensional spaces. As discussed in the introduction, maximisation of the acquisition function in a crucial step when working with unknown high dimensional search spaces. Given the same computation budget, the larger the expanded search space, the less accurate the maximiser suggested by the acquisition step is. To improve this step, our solution is to restrict the search space. We propose a novel volume expansion strategy as follows. Starting from $\mathcal{X}_0$, the search space $\mathcal{H}_t$ at iteration $t$ with $t \geq 1$ is defined as

$$\mathcal{H}_t = \{H(\mathbf{z}_t^1, l_h) \cup ... \cup H(\mathbf{z}_t^{N_t}, l_h)\} \cap \mathcal{X}_t, \tag{3}$$

where $\mathcal{X}_t$ is the search space of HuBO and is defined in 2, $H(\mathbf{z}_t^i, l_h)$ is a $d$-dimensional hypercube centered at $\mathbf{z}_t^i$ with size $l_h$, and $N_t$ denotes the number of such hypercubes at iteration $t$. To handle the computational requirement, we choose $l_h$ to be small. Thus, at an iteration $t$, we maximise the acquisition function on only this finite set of hypercubes in $\mathcal{X}_t$ with small size.

Importantly, we can show that the maximisation on such hypercubes can result in low regret by proposing a strategy to choose the set of hypercubes. Formally, at iteration $t$ we choose $N_t = N_0\lceil t^\lambda \rceil$ where $\lambda \geq 0$, $N_0 \in \mathbb{N}$ and $N_0 \geq 1$. We choose $N_t$ hypercubes with centres $\{\mathbf{z}_t^i\}$ which are sampled uniformly at random from $\mathcal{X}_t$. We refer to this algorithm as HD-HuBO which is described in Algorithm 2. We use the acquisition function $u_t(\mathbf{x})$ with $\beta_t = 2log(\pi^2 t^2/\delta) + 2dlog(2s_2 l_h d\sqrt{log(6ds_1/\delta)}t^2)$, where $s_1, s_2$ is defined in Theorem 4.

### 5.1   Convergence Analysis for HD-HuBO Algorithm

In this section, we analyse the convergence of our proposed HD-HuBO algorithm. Similar to HuBO algorithm, a Reachability property is necessary to guarantee the convergence. On the restricted search space $\mathcal{H}_t$, it is a crucial challenge. To overcome this, we estimate the distance between $\mathbf{x}^*$ and $\mathbf{x}_t^*$ which is the closest point to $\mathbf{x}^*$ in the search space $\mathcal{H}_t$, as shown in the following Theorem 3. Thus, although we cannot maintain the Reachability property as in HuBO algorithm, we can still obtain a similar Reachability property with high probability if $\lambda > d(\alpha + 1)$ and $-1 \leq \alpha < 0$, where $\alpha$ is the expansion rate of the search space and is defined as in HuBO algorithm.

**Theorem 3.**  Pick a $\delta \in (0, 1)$. Let $\mathbf{x}_t^* \in \mathcal{H}_t$ be the closest point to $\mathbf{x}^*$ in the search space $\mathcal{H}_t$. For any $t > T_0$ and $-1 \leq \alpha < 0$, with probability greater than $1 - \delta$, we have

$$||\mathbf{x}_t^* - \mathbf{x}^*||_2 < \frac{2(b-a)}{\pi}(\Gamma(\frac{d}{2}+1))^{\frac{1}{d}}(log(\frac{1}{\delta}))^{\frac{1}{d}} M_t, \tag{4}$$

where the constant $T_0$ is defined in Theorem 1, $\Gamma$ is the gamma function, and $M_t = (2 + ln(t))t^{-\frac{\lambda}{d}}$ if $\alpha = -1$, otherwise, $M_t = 2(\alpha + 1)^{-1}t^{-\frac{\lambda}{d}}$ if $-1 < \alpha < 0$.

By Theorem 3, for both cases $\alpha = -1$ and $-1 < \alpha < 0$, $\lim_{t \to \infty} M_t \to 0$ if $\lambda > d(\alpha + 1)$. Therefore, $\lim_{t \to \infty} ||\mathbf{x}_t^* - \mathbf{x}^*||_2 \to 0$ if $\lambda > d(\alpha + 1)$. The Reachability property is guaranteed with high probability.

Using Theorem 3, we derive a cumulative regret for our proposed HD-HuBO algorithm as follows.

**Theorem 4** (Cumulative Regret $R_T$ of HD-HuBO Algorithm). Let $f \sim \mathcal{GP}(\mathbf{0}, k)$ with a stationary covariance function $k$. Assume that there exist constants $s_1, s_2 > 0$ such that $\mathbb{P}[\sup_{\mathbf{x} \in \mathcal{X}} |\partial f/\partial x_i| > L] \le s_1 e^{-(L/s_2)^2}$ for all $L > 0$ and for all $i \in \{1, 2, ..., d\}$. Pick a $\delta \in (0, 1)$. Then, with $T > T_0$, under conditions $\lambda > d(\alpha + 1)$, $-1 \le \alpha < 0$, $l_h > 0$, the cumulative regret of proposed HD-HuBO algorithm is bounded as

- $R_T \le \mathcal{O}^*(T^{\frac{(\alpha+1)d+1}{2}} + (log(\frac{6}{\delta}))^{\frac{1}{d}} B_T)$ if $k$ is a SE kernel,

- $R_T \le \mathcal{O}^*(T^{\frac{d^2(\alpha+2)+d}{4\nu+2d(d+1)}+\frac{1}{2}} + (log(\frac{6}{\delta}))^{\frac{1}{d}} B_T)$ if $k$ is a Matérn kernel,

with probability greater than $1 - \delta$, where $B_T = U_T V_T$, where $U_T = 2 + ln(T)$ if $\alpha = -1$, otherwise $U_T = 2(\alpha + 1)^{-1}$, and $V_T = 1 + ln(T)$ if $\lambda = d$, otherwise $V_T = 1 + \frac{d}{d-\lambda} \max\{1, T^{1-\frac{\lambda}{d}}\}$.

**Sub-linear Regret** The upper bound on $R_T$ is sub-linear because we have $\lim_{T \to \infty} \frac{B_T}{T} = 0$ for all the cases of $U_T$ and $V_T$. The conditions on $\alpha$ is maintained as in HuBO to guarantee the sub-linear regret for SE kernels and Matérn kernels. Together with conditions on $\lambda$, our proposed HD-HuBO obtains a sub-linear cumulative regret for SE kernels if $-1 \le \alpha < -1 + \frac{1}{d}$ and $\lambda > d(\alpha + 1)$, and for Matérn kernels if $-1 \le \alpha < -1 + \frac{2\nu}{d^2}$ and $\lambda > d(\alpha + 1)$. We note that the regret bound of HD-HuBO is higher than HuBO's regret bound, however HD-HuBO uses only the restricted search space of HuBO.

## 6 Discussion

**On the use of hypercubes** While Eriksson et al. [6] proposed to use hypercubes for high dimensional BO, our main contribution is a high dimensional BO for *unknown search spaces* (a novel problem setting). Unlike in [6] where the number of hypercubes are fixed, we provide a rigorous method to increase the number of hypercubes with iterations which is required for our case as the search space is unknown and an initial randomly specified search space needs to keep growing to ensure the convergence. Further, in [6], there is no theoretical analysis of regret, nor there is any rigorous analysis on the number of hypercubes.

**On the effect of $\alpha$ and $\lambda$ parameters** For both our algorithms, to achieve the tightest sub-linear term in the regret, the parameter $\alpha$ needs to be as small as possible while being in the kernel-specific permissible range. However, a small $\alpha$ may lead to a higher value of $T_0$, which may increase finite-time regret. For HD-HuBO, a high $\lambda$ may lead to a larger volume of the restricted search space. Our algorithm offers a range of operating choices (through the choice of $\lambda$) while still guaranteeing different grades of sub-linear rate.

## 7 Experiments

To evaluate the performance of our algorithms, HuBO and HD-HuBO, we have conducted a set of experiments involving optimisation of five benchmark functions and three real applications. We compare our algorithms against five baselines: (1) UBO: the method in a recent paper [8], (2) FBO: the method in [18], (3) Vol2: BO with the search space volume doubled every $3d$ iterations [23], (4) Re-H: the Regularized acquisition function with a hinge-quadratic prior [23]; (5) Re-Q: the Regularized acquisition function with a quadratic prior [23].

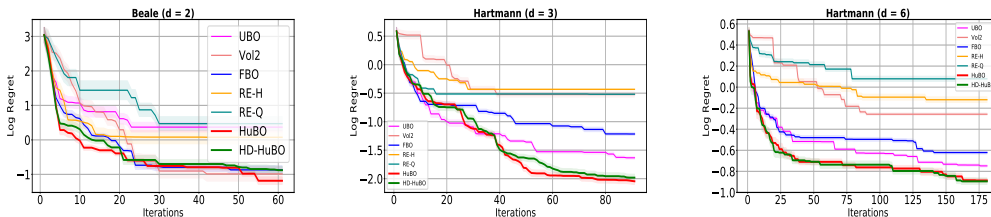

Figure 1: Comparison of baselines and the proposed methods in low dimensions.

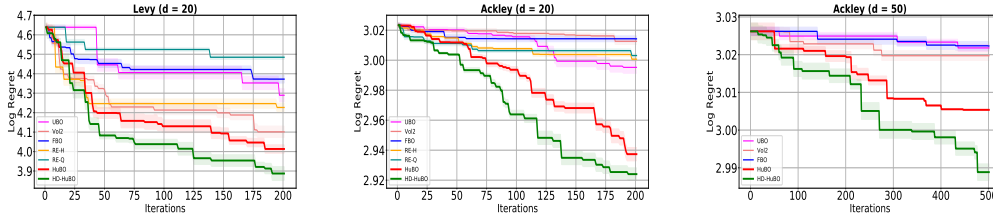

Figure 2: Comparison of baselines and the proposed methods in high dimensions.

**Experimental settings**    Following the setting of the initial search space $\mathcal{X}_0$ as in all baselines [23, 18, 8], we select the $\mathcal{X}_0$ as 20% of the pre-defined function domain. For example, if $\mathcal{X} = [0,1]^d$, the size of $\mathcal{X}_0$ is the 0.2 where its center is placed randomly in the domain $[0,1]^d$. For our algorithms, we set $\mathcal{C}_{inital}$ as 10 times to the size of $\mathcal{X}_0$ along each dimension. We note that we also validate our algorithms by considering additionally a case where $\mathcal{X}_0$ is only 2% (very small) of the pre-defined function domain. We report this case in the **supplementary material**.

For all algorithms, the Squared Exponential kernel is used to model GP. The GP models are fitted using the Maximum Likelihood Estimation. The function evaluation budget is set to $30d$ in low dimensions and $10d$ in high dimensions where $d$ is the input dimension. The experiments were repeated 15 times and average performance is reported. For the error bars (or variances), we use the standard error: Std. Err = Std. Dev$/\sqrt{n}$, $n$ being the number of runs. Following our theoretical

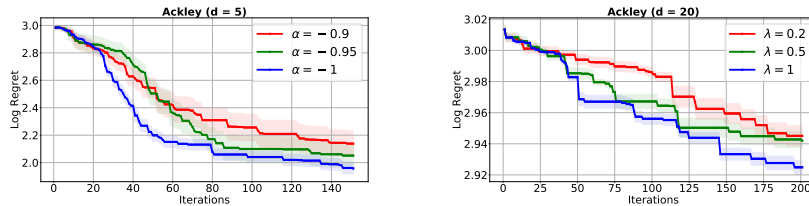

Figure 3: The study of $\alpha$ (for HuBO) and $\lambda$ (for HD-HuBO) in terms of best function value vs iterations. **Left**: Ackley function (d = 5) and different values of $\alpha$: -0.9; -0.95; -1. **Right**: Ackley function (d =20) with $\alpha = -1$ and different values of $\lambda$: 0.2; 0.5; 1.

results, we choose $\alpha = -1$. By this way, any $\lambda > 0$ is valid as per our theoretical results and more importantly, it allows to minimize the number of hypercubes in HD-HuBO algorithm, and thus reduces the computations. We use $\lambda = 1, N_0 = 1$ and thus $N_t = t$. This means that at iteration $t$, we use $t$ hypercubes for the maximisation of acquisition function. We set the size of hypercubes, $l_h$ as 10% of $\mathcal{X}_0$. All algorithms are given an equal computational budget to maximise acquisition functions. We also report the average time with each test function in the **supplementary material** (see Table 1).

### 7.1    Optimisation of Benchmark Functions

We test the algorithms on several benchmark functions: Beale, Hartmann3, Hartmann6, Ackley, Levy functions. We evaluate the progress of each algorithm using the log distance to the true optimum, that is, $\log_{10}(f(\mathbf{x}^*) - f^+)$ where $f^+$ is the best function value found so far. For each test function, we repeat the experiments 15 times. We plot the mean and a confidence bound of one standard deviation across all the runs. Results are reported in Figure 1 for low dimensions and in Figure 2 for higher dimensions. From Figure 1, we can see that Vol2, Re-H and Re-Q perform poorly in most cases. We note that there is no convergence guarantee for methods Vol2, Re-H and Re-Q. Our HuBO method outperforms baselines. In high dimensions, both HuBO and HD-HuBO outperform baselines. Particularly, since the maximisation of the acquisition function is performed on a restricted search space compared to HuBO, the performance of HD-HuBO is notable.

We would like to emphasise that, different from traditional BO algorithms with fixed search space where error bars (or variances) of regret curves tend to get tighter over time, in the context of unbounded search space where the search space is being expanded over time, error bars do not always have this property, they may even become higher over time till the search spaces have not contained

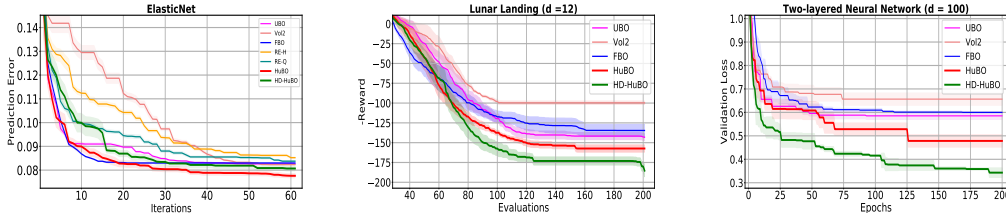

Figure 4: **Left**: Prediction accuracy vs Iterations for MNIST dataset using Elastic Net. **Middle**: Rewards vs Evaluations for 12D Lunar lander. **Right**: Validation loss vs Epochs for learning parameters of a Two-layered Neural Network.

the global optimum. This trend can be seen for many unbounded search space methods such as in [8] and [18] in our references.

**On the Expansion Rate $\alpha$ and Number of Hypercubes $\lambda$** The space expansion rate $\alpha$ and the number of hypercubes are control parameters in our method. For SE kernels, $-1 \leq \alpha < -1 + \frac{1}{d}$ is needed. However, in high dimensions, this range is tight. Therefore, to test the effect of $\alpha$, we consider HuBO with low dimensions. We create many variants of HuBO ($\alpha = -1, \alpha = -0.95$ and $\alpha = -0.9$). As a testbed, we use 5-dim Ackley function. Figure 3 shows that smaller values of $\alpha$ performs better achieving tighter regrets. To study the effect of $\lambda$, we fix $\alpha = -1$ and create three variants of HD-HuBO using $\lambda = 0.2, \lambda = 0.5$ and $\lambda = 1$. We observe that larger values of $\lambda$ achieve tighter regrets. These results validate our theoretical analysis.

## 7.2 Applications to Machine Learning Models

**Elastic Net** Elastic net is a regression method that has the $L_1$ and $L_2$ regularization parameters. We tune $w_1$ and $w_2$ where $w_1 > 0$ expresses the magnitude of the regularisation penalty while $w_2 \in [0, 1]$ expresses the ratio between the two penalties. We tune $w_1$ in the normal space while $w_2$ is tuned in an exponent space (base 10). The $\mathcal{X}_0$ is randomly placed box in the domain $[0, 1] \times [-3, -1]$. We implement the Elastic net model by using the function SGDClassifier in the scikit-learn package [20]. We train the model using the MNIST train dataset and then evaluate the model using the MNIST test dataset. Bayesian optimisation method suggests a new hyperparameter setting based on the prediction accuracy on the test set. As seen from Figure 4 (Left), HuBO performs better than the baselines. In low dimensions, the restriction of the search space in HD-HuBO can influence the efficiency of BO, thus it is less efficient than HuBO.

**Lunar Landing Reinforcement Learning** In this task, the goal is to learn a controller for a lunar lander. The state space for the lunar lander is the position, angle, time derivatives, and whether or not either leg is in contact with the ground. The objective is to maximize the average final reward. The controller we learn is a modification of the original heuristic controller where there are twelve design parameters [6]. Each design parameter is tuned heuristically between $[0, 2]$. We follow their setting however instead of assuming a fixed search space as $[0, 2]^{12}$, we assume that the search space is unknown. Thus, $\mathcal{X}_0$ is randomly placed in the domain $[0, 2]^{12}$. We set $\alpha = -1$ and $\lambda = 1$. Our methods HuBO and HD-HuBO eventually learn the best controllers although at early iterations,

**Parameter Tuning for Machine Learning Models** We evaluate the algorithms on a two-layered neural network parameter optimisation task. Here we are given a CNN with one hidden layer of size 10. We denote the weights between the input and the hidden layer by $W_1$ and the weights between the hidden and the output layer by $W_2$. The goal is to find the weights that minimize the loss on the MNIST data set. We optimize $W_2$ by BO methods while $W_1$ are optimized by Adam algorithm. We choose the same network architecture as used by [19, 26], however different from them, we assume that the search space is unknown. We randomly choose an initial box in the domain $[0, \sqrt{d}]^{100}$. We compare our methods to UBO, Vol2, FBO using the validation loss. We set $\alpha = -1$ and $\lambda = 0.5$ for our methods. Both our methods outperform all the baselines, especially, HD-HuBO.

## 8 Conclusion

We propose a novel BO algorithm for global optimisation in an unknown search space setting. Starting from a randomly initialised search space, the search space shifts and expands as per a hyperharmornic series. The algorithm is shown to efficiently converge to the global optimum. We

extend this algorithm to high dimensions, where the search space is restricted on a finite set of small hypercubes so that maximisation of the acquisition function is efficient. Both algorithms are shown to converge with sub-linear regret rates. Application to many optimisation tasks reveals the better sample efficiency of our algorithms compared to the existing methods.

## Broader Impact Statement

This work has potential to enable the scientists and researchers from the experimental design community to optimise the design of products and processes without the need to specify a search space, which is usually not known accurately when pursuing new products and processes. There are no unethical side of this research or any ill-effects on society.

## Acknowledgments

This research was partially funded by the Australian Government through the Australian Research Council (ARC). Prof Venkatesh is the recipient of an ARC Australian Laureate Fellowship (FL170100006).

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
