[Supplementary Material]

# Sub-linear Regret Bounds for Bayesian Optimisation in Unknown Search Spaces - Supplementary Material

In section 1, we first provide some auxiliary results which facilitate the proofs. We present the proofs of Theorem 1, Theorem 2, Theorem 3 and Theorem 4 in next sections. Finally, we provide additional benchmarking results in section 6.

## 1 Auxiliary Results

### 1.1 Properties of the Volume Expansion Strategy

**Lemma 1.** For every $t \geq 1$, the search space $\mathcal{X}_t$ has the $[a_t, b_t]^d$ form where $b_t - a_t = (b - a)(1 + \sum_{j=1}^{t} j^\alpha)$.

*Proof.* We prove the statement by induction. If $t = 1$ then by definition of the transformation in section 4, $X_1' = [a_1', b_1']^d$ where $a_1' = a_0 - \frac{b-a}{2} 1^\alpha$ and $b_1' = b_0 + \frac{b-a}{2} 1^\alpha$. Hence, $b_1' - a_1' = 2(b-a)$. By definition of the transformation $X_1' \to \mathcal{X}_1$, the size and the form of $\mathcal{X}_1$ is preserved from $\mathcal{X}_1'$. Therefore $\mathcal{X}_1 = [a_1, b_1]^d$ and $b_1 - a_1 = 2(b-a)$.

We assume that the statement is true for $t \geq 1$. We consider the transformation $\mathcal{X}_t \to \mathcal{X}_{t+1}' \to \mathcal{X}_{t+1}$. We have $a_{t+1}' = a_t - \frac{b-a}{2}(t+1)^\alpha$ and $b_{t+1}' = b_t + \frac{b-a}{2}(t+1)^\alpha$. Hence, $b_{t+1}' - a_{t+1}' = b_t - a_t + (b-a)(t+1)^\alpha$. By the inductive hypothesis, we have $\mathcal{X}_t = [a_t, b_t]^d$ and $b_t - a_t = (b-a)(1 + \sum_{j=1}^{t} j^\alpha)$. Therefore, $b_{t+1}' - a_{t+1}' = (b-a)(1 + \sum_{j=1}^{t+1} j^\alpha)$. By the transformation, the size and the form of $\mathcal{X}_{t+1}$ is preserved from $\mathcal{X}_{t+1}'$. Thus, $\mathcal{X}_{t+1} = [a_{t+1}, b_{t+1}]^d$ where $b_{t+1} - a_{t+1} = (b-a)(1 + \sum_{j=1}^{t+1} j^\alpha)$. The statement holds for any $t \geq 1$. □

Given a finite domain $\mathcal{X}$, we denote the volume of $\mathcal{X}$ by $Vol(\mathcal{X})$.

**Lemma 2.** For every horizon $T > 0$, set $\mathcal{C}_T = [c_{min} - (b-a)(1 + \sum_{j=1}^{T} j^\alpha)/2, c_{max} + (b-a)(1 + \sum_{j=1}^{T} j^\alpha)/2]^d$. Then for every $1 \leq t \leq T$, $X_t \subseteq \mathcal{C}_T$.

*Proof.* We also prove this statement by induction. If $T = 1$ then by Lemma 1, $\mathcal{X}_1 = [a_1, b_1]^d$ where $b_1 - a_1 = 2(b-a)$. By the transformation, the center of $\mathcal{X}_1$ only moves in the domain $\mathcal{C}_{initial}$. If we set $\mathcal{C}_T = [c_{min} - (b-a), c_{max} + (b-a)]^d$ then $\mathcal{X}_1 \subseteq \mathcal{C}_T$.

We assume that the statement is true for $T \geq 1$. By the inductive hypothesis, for every $1 \leq t \leq T$, $X_t \subseteq \mathcal{C}_T = [c_{min} - (b-a)(1 + \sum_{j=1}^{T} j^\alpha)/2, c_{max} + (b-a)(1 + \sum_{j=1}^{T} j^\alpha)/2]^d$. We set $\mathcal{C}_{T+1} = [c_{min} - (b-a)(1 + \sum_{j=1}^{T+1} j^\alpha)/2, c_{max} + (b-a)(1 + \sum_{j=1}^{T+1} j^\alpha)/2]^d$. First, we have $\mathcal{C}_T \subset \mathcal{C}_{T+1}$. Next we prove that $\mathcal{X}_{T+1} \subseteq \mathcal{C}_{T+1}$. Indeed, by Lemma 1, $X_{T+1} = [a_{T+1}, b_{T+1}]^d$ where $b_{T+1} - a_{T+1} = (b-a)(1 + \sum_{j=1}^{T+1} j^\alpha)$. By the transformation, the center of $\mathcal{X}_{T+1}$ only moves in the domain $\mathcal{C}_{initial}$. It implies that $\mathcal{X}_{T+1}$ belongs to $\mathcal{C}_{T+1}$. The statement holds for any $T \geq 1$. □

## 1.2 Properties of The Gamma Function and The Hyperharmonic Series

**Lemma 3.** (Lower bounds of a partial sum of a hyperharmonic series,[2]) Given a partial sum of a hyperharmonic series $p_n = \sum_{j=1}^n j^\alpha$, where $n \in \mathbb{N}$. Then,

- $p_n > \frac{(n+1)^{\alpha+1}-1}{\alpha+1}$ if $-1 \le \alpha < 0$,

- $p_n > ln(n+1)$ if $\alpha = -1$.

**Lemma 4.** (Upper Bounds of a Hyperharmonic Series, [2]) Given a hyperharmonic series $p_n = \sum_{j=1}^n j^\alpha$, where $n \in \mathbb{N}$. Then,

- $p_n < 1 + \frac{n^{1+\alpha}-1}{1+\alpha}$ if $-1 \le \alpha < 0$,

- $p_n < 1 + ln(n)$ if $\alpha = -1$

**Lemma 5.** (Bounding $p$-series when $p > 1$, [1]) Given a $p$-series $s_n = \sum_{k=1}^n \frac{1}{k^p}$, where $n \in \mathbb{N}$. Then,

$$s_n < \zeta(p) < \frac{1}{p-1} + 1$$

for any $n$, where $\zeta(p) = \sum_{k=1}^\infty \frac{1}{k^p}$ is Euler–Riemann zeta function that always converges. For example, $\zeta(3/2) \approx 2.61, \zeta(2) = \frac{\pi^2}{6}$.

**Lemma 6.** $\Gamma(\frac{d}{2}+1)^{\frac{1}{d}} < \sqrt{d+2}$

*Proof.* We consider two cases:

- if $d = 2n$, where $n \in \mathbb{N}$ then $\Gamma(\frac{d}{2}+1) = \Gamma(n+1) = n!$

- if $d = 2n+1$, where $n \in \mathbb{N}$ then $\Gamma(\frac{d}{2}+1) = \Gamma(n+1+\frac{1}{2}) = n!\Gamma(\frac{1}{2}) = \sqrt{\pi}n! < 2n!$

Hence, in both case, $\Gamma(\frac{d}{2}+1) < 2n!$. By Cauchy-Schwarz, we have:$n! < (\frac{1+2+...+n)}{n})^n = (\frac{n+1}{2})^n$. However, $n \le \frac{d}{2}$. Thus, $(\Gamma(\frac{d}{2}+1))^{\frac{1}{d}} < 2(\frac{n+1}{2})^{\frac{n}{d}} < \sqrt{2(n+1)} < \sqrt{d+2}$. $\square$

## 2 Proof of Theorem 1

**Theorem 1** (Reachability). If $\alpha \ge -1$, then the HuBO algorithm guarantees that there exists a constant $T_0 > 0$ (independent of $t$) such that when $t > T_0$, $\mathcal{X}_t$ contains $\mathbf{x}^*$.

We denote the center of the user-defined finite region $\mathcal{C}_{initial} = [c_{min}, c_{max}]^d$ as $\mathbf{c}_0$. By the assumption of $\mathbf{x}^*$ being not at infinity, there exists a smallest range $[a_g, b_g]^d$ so that both $\mathbf{x}^*$ and $\mathbf{c}_0$ belong to $[a_g, b_g]^d$. By induction, the search space $\mathcal{X}_t$ at iteration $t$ is a hypercube, denoted by $[a_t, b_t]^d$. Following our search space expansion, the center of $\mathcal{X}_t$ only moves in region $\mathcal{C}_{initial}$. Therefore, for each dimension $i$, we have in the worst case, $(b_t - [\mathbf{c}_0]_i)$ is at least $\frac{c_{min}-c_{max}}{2} + \frac{b_t-a_t}{2}$ and $([\mathbf{c}_0]_i - a_t)$ is at most $\frac{c_{max}-c_{min}}{2} - \frac{b_t-a_t}{2}$. By Lemma 1, the size of $\mathcal{X}_t$ as $b_t - a_t : b_t - a_t = (b-a)(1+\sum_{j=1}^t j^\alpha)$. Therefore, $(b_t - [\mathbf{c}_0]_i)$ is at least $\frac{c_{min}-c_{max}}{2} + \frac{b-a}{2}(1 + \sum_{j=1}^t j^\alpha)$ and $([\mathbf{c}_0]_i - a_t)$ is at most $\frac{c_{max}-c_{min}}{2} - \frac{b-a}{2}(1 + \sum_{j=1}^t j^\alpha)$.

If there exists a $T_0$ such that two conditions satisfy: (1) $\frac{c_{min}-c_{max}}{2} + \frac{b-a}{2}(1 + \sum_{j=1}^{T_0} j^\alpha) \ge b_g$, and (2) $\frac{c_{max}-c_{min}}{2} - \frac{b-a}{2}(1 + \sum_{j=1}^{T_0} j^\alpha) < a_g$, then we can guarantee that for all $t > T_0$, the search space $\mathcal{X}_t$ will contain $[a_g, b_g]^d$ and thus also contain $\mathbf{x}^*$.

Such a $T_0$ exists because $p_t = \sum_{j=1}^t j^\alpha$ is a diverging sum with $t$ when $\alpha \ge -1$. Indeed, by Lemma 3, we have

- $p_t > \frac{(t+1)^{\alpha+1}-1}{\alpha+1}$ if $-1 \le \alpha < 0$,

- $p_t > ln(t+1)$ if $\alpha = -1$.

For both cases, $\lim_{t\to\infty} p_t \to \infty$. From the conditions (1) and (2), we can see that $T_0$ is a function of parameters $a$, $b$, $c_{min}$, $c_{max}$, $\alpha$ and $a_g, b_g$. Since $a$, $b$, $c_{min}$, $c_{max}$ and $\alpha$ are determined at the beginning of the HuBO algorithm and do not change, such a constant (although unknown) $T_0$ exists.

## 3   Proof of Theorem 2

To derive an upper bound of the cumulative regret of the HuBO algorithm for SE kernels and Matérn kernels, we first derive an upper bound of the cumulative regret for a general class of kernels according to the maximum information gain. We do this in the following Proposition 1. Next, we provide upper bounds for the maximum information gain on SE kernels and Matérn kernels. We do that in Proposition 2. Finally, we prove the correctness of Theorem 2 by combining Proposition 1 and Proposition 2.

**Proposition 1.** Let $f \sim \mathcal{GP}(\mathbf{0}, k)$ with a stationary covariance function $k$. Assume that $-1 \leq \alpha < 0$ and there exist constants $s_1, s_2 > 0$ such that $\mathbb{P}[sup_{\mathbf{x}\in\mathcal{X}}|\partial f/\partial x_i| > L] \leq s_1 e^{-(L/s_2)^2}$ for all $L > 0$ and for all $i \in \{1, 2, ..., d\}$. Pick a $\delta \in (0, 1)$. Set $\beta_T = 2log(4\pi_t/\delta) + 4dlog(dTs_2(b - a)(1 + \sum_{j=1}^{T} j^\alpha)\sqrt{log(4ds_1/\delta)})$. Thus, there is a constant $C$ such that for any horizon $T > T_0$, the cumulative regret of the proposed HuBO algorithm is bounded as

$$R_T \leq C + \sqrt{C_1 T \beta_T \gamma_T(\mathcal{C}_T)} + \frac{\pi^2}{6}$$

, with probability $1 - \delta$, where the domain $\mathcal{C}_T = [c_{min} - (b - a)(1 + \sum_{j=1}^{T} j^\alpha)/2, c_{max} + (b - a)(1 + \sum_{j=1}^{T} j^\alpha)/2]^d$, and $\gamma_T(\mathcal{C}_T)$ is the maximum information gain for any $T$ observations in the domain $\mathcal{C}_T$ (see [4]).

*Proof.* Let us denote by $f_t^*$ the optimum in the search space $\mathcal{X}_t$, and denote by $g_t$ the gap between the global optimum and the optimum in the search space $\mathcal{X}_t$. Formally, $g_t = f(x^*) - f_t^*$. We consider two cases:

- $1 \leq t \leq T_0$. Then with the probability $1 - \delta$, $r_t$ can be bounded as follows:

$$
\begin{align}
r_t &= f(x^*) - f(x_t) \tag{1}\\
&= f_t^* - f(x_t) + g_t \tag{2}\\
&= f_t^* - \mu_{t-1}(x^*) + \mu_{t-1}(x^*) - f(x_t) + g_t \tag{3}\\
&\leq f(x^*) - \mu_{t-1}(x^*) + \mu_{t-1}(x^*) - f(x_t) + g_t \tag{4}\\
&\leq \sqrt{\beta_t}\sigma_{t-1}(x^*) + \mu_{t-1}(x^*) - f(x_t) + g_t \tag{5}\\
&\leq \sqrt{\beta_t}\sigma_{t-1}(x_t) + \mu_{t-1}(x_t) - f(x_t) + g_t \tag{6}\\
&\leq 2\sqrt{\beta_t}\sigma_{t-1}(x_t) + g_t \tag{7}
\end{align}
$$

  where the inequality (4) holds as $f_t^* \leq f(x^*)$, the inequality (5) holds as $f(x^*) \leq \mu_{t-1}(x^*) + \sqrt{\beta_t}\sigma_{t-1}(x^*)$ with probability $1 - \delta$ ( the proof is similar to Lemma 5.5 of [4]), the inequality (6) holds as $\sqrt{\beta_t}\sigma_{t-1}(x^*) + \mu_{t-1}(x^*) = \mu_t(x^*) \leq \sqrt{\beta_t}\sigma_{t-1}(x_t) + \mu_{t-1}(x_t) = \mu_t(x_t)$ ( recall that $x_t = \text{argmax}_{x\in\mathcal{X}_t} u_t(x)$), and finally inequality (7) holds as $\mu_{t-1}(x_t) - \sqrt{\beta_t}\sigma_{t-1}(x_t) \leq f(x_t)$ with probability $1 - \delta$ ( the proof is similar to Lemma 5.1 of [4]).

- $t > T_0$. By Theorem 1, the search space $\mathcal{X}_t$ contains $x^*$. Similar to the idea of [4], we can use a set of *discretizations* of $\mathcal{X}_t$ to achieve a valid confidence interval on $x^*$. By proof similar to Lemma 5.8 of [4], we achieve: $r_t \leq 2\sqrt{\beta_t}\sigma_{t-1}(x_t) + \frac{1}{t^2}$.

Combining the two cases, we achieve $R_T = \sum_{t=1}^{T} r_t \leq \sum_{t=1}^{T_0} g_t + 2\sum_{t=1}^{T} \sqrt{\beta_t}\sigma_{t-1}(x_t) + \sum_{t=T_0+1}^{T} \frac{1}{t^2} \leq C + 2\sum_{t=1}^{T} \sqrt{\beta_t}\sigma_{t-1}(x_t) + \sum_{t=1}^{T} \frac{1}{t^2} \leq C + 2\sum_{t=1}^{T} \sqrt{\beta_t}\sigma_{t-1}(x_t) + \sum_{t=1}^{T} \frac{1}{t^2} + \frac{\pi^2}{6}$, where we set $C = \sum_{t=1}^{T_0} g_t$. To make our problem in context of unknown search spaces tractable, we

assume that the function $f$ is *finite* on any finite domain of $\mathbb{R}^d$. It implies that for every $1 \leq t \leq T_0$, $g_t$ is finite. Further, by definition of $T_0$, $T_0$ is the constant and independent of $T$. Thus, $C$ is also a constant and is independent of $T$.

Next, we derive an upper bound on $\sum_{t=1}^{T} \sqrt{\beta_t}\sigma_{t-1}(x_t)$. By Lemma 2, for every $1 \leq t \leq T$, $\mathcal{X}_t \subseteq \mathcal{C}_T$. Similar to the proof of Lemma 5.4 of [4] we can achieve

$$\sum_{t=1}^{T} 4\beta_t \sigma_{t-1}^2(x_t) \leq C_1 \beta_T \gamma_T(\mathcal{C}_T), \tag{8}$$

where $C_1 = 8/log(1+\sigma^2)$, $\beta_t = 2log(4\pi_t/\delta) + 4dlog(dts_2(b-a)(1+\sum_{j=1}^{t}j^\alpha)\sqrt{log(4ds_1/\delta)})$.

By Cauchy-Schwarz, we have:

$$\sum_{t=1}^{T} \sqrt{\beta_t}\sigma_{t-1}(x_t) \leq \sqrt{C_1 T \beta_T \gamma_T(\mathcal{C}_T)} \tag{9}$$

Therefore, $R_T \leq C + \sqrt{C_1 T \beta_T \gamma_T(\mathcal{C}_T)} + \frac{\pi^2}{6}$. $\qquad\square$

**Proposition 2.** We assume the kernel function $k$ satisfies $k(x, x') \leq 1$. Then,

- For SE kernels: $\gamma_T(\mathcal{C}_T) = \mathcal{O}(T^{(\alpha+1)d})$,

- For Matérn kernels with $\nu > 1$: $\gamma_T(\mathcal{C}_T) = \mathcal{O}(T^{\frac{d^2(\alpha+2)+d}{2\nu+d(d+1)}})$

*Proof.* For SE kernels, by the proof similar as in Theorem 5 of [4], we can bound $\gamma_T(\mathcal{C}_T)$ as $\gamma_T(\mathcal{C}_T) \leq \mathcal{O}(Vol(\mathcal{C}_T)log(T))$. By definition, $\mathcal{C}_T = [c_{min} - (b-a)(1+\sum_{j=1}^{T}j^\alpha)/2, c_{max} + (b-a)(1+\sum_{j=1}^{T}j^\alpha)/2]^d$. Hence, $Vol(\mathcal{C}_T) = (c_{max} - c_{min} + (b-a)(1+\sum_{j=1}^{T}j^\alpha))^d$. We consider two cases on $\alpha$:

- $\alpha = -1$. By Lemma 4, $\sum_{j=1}^{T}j^\alpha < 1 + ln(T)$. Hence, $Vol(\mathcal{C}_T) < (c_{max} - c_{min} + (b-a)(2 + ln(T)))^d$. Therefore, $\gamma_T(\mathcal{C}_T) \leq \mathcal{O}((ln(T))^{d+1})$.

- if $-1 < \alpha < 0$. By Lemma 4, $\sum_{j=1}^{T}j^\alpha < 1 + \frac{T^{1+\alpha}-1}{1+\alpha}$. Hence, $Vol(\mathcal{C}_T) = (c_{max} - c_{min} + (b-a)(1+\sum_{j=1}^{T}j^\alpha))^d < (c_{max} - c_{min} + \frac{b-a}{(1+\alpha)^d}(2\alpha + 1 + T^{\alpha+1}))^d$. Thus, $Vol(\mathcal{C}_T) = \mathcal{O}(T^{(\alpha+1)d})$.

Thus, $\gamma_T(\mathcal{C}_T) = \mathcal{O}(T^{(\alpha+1)d})$.

For Matérn kernels, by the proof similar as in Theorem 5 of [4], we can bound $\gamma_T(\mathcal{C}_T)$ as $\gamma_T(\mathcal{C}_T) = \mathcal{O}(T_* log(Tn_T))$, where $n_T = 2Vol(\mathcal{C}_T)(2\tau + 1)T^\tau(logT)$ and $T_* = (Tn_T)^{d/(2\nu+d)}(log(Tn_T))^{-d/(2\nu+d)}$, $\tau$ is a parameter. We consider two cases:

- if $\alpha = -1$, $\sum_{j=1}^{T}j^\alpha < 1 + ln(T)$. We have $Vol(\mathcal{C}_T) = \mathcal{O}(logT)$ and $\mathcal{O}(T_* log(Tn_T)) = \mathcal{O}(T^{\frac{(\tau+1)d}{2\nu+d}}(logT))$. We choose $\tau = \frac{2\nu d}{2\nu+d(d+1)}$ to match this term with $\mathcal{O}(T^{1-\frac{\tau}{d}})$. Thus, $\gamma_T(\mathcal{C}_T) = \mathcal{O}(T^{1-\frac{\tau}{d}}) = \mathcal{O}(T^{\frac{d(d+1)}{2\nu+d(d+1)}})$.

- if $-1 < \alpha < 0$, we obtain $Vol(\mathcal{C}_T) = \mathcal{O}(T^{(\alpha+1)d})$. Thus, $\mathcal{O}(T_* log(Tn_T)) = \mathcal{O}(T^{\frac{((\alpha+1)d+\tau+1)d}{2\nu+d}}(logT))$. To match this term with $\mathcal{O}(T^{1-\frac{\tau}{d}})$ we choose $\tau$ such that:

$$\frac{((\alpha+1)d+\tau+1)d}{2\nu+d} = 1 - \frac{\tau}{d}$$

This is equivalent to $\tau = \frac{2\nu d - d^3(\alpha+1)}{2\nu+d(d+1)}$. Thus, $\gamma_T(\mathcal{C}_T) = \mathcal{O}(T^{1-\frac{\tau}{d}}) = \mathcal{O}(T^{\frac{d^2(\alpha+2)+d}{2\nu+d(d+1)}})$.

Since, when $\alpha = -1$, $\gamma_T(\mathcal{C}_T) = \mathcal{O}(T^{\frac{d2(\alpha+2)+d}{2\nu+d(d+1)}}) = \mathcal{O}(T^{\frac{d(d+1)}{2\nu+d(d+1)}})$. Thus, we can write $\gamma_T(\mathcal{C}_T) = \mathcal{O}(T^{\frac{d2(\alpha+2)+d}{2\nu+d(d+1)}})$ for $-1 \leq \alpha < 0$. $\qquad\qquad\qquad\qquad\qquad\qquad\qquad\qquad\qquad\qquad\qquad\qquad\qquad\square$

Combining Proposition 1 and Proposition 2, we achieve Theorem 2.

**Theorem 2** (Cumulative Regret $R_T$ of HuBO Algorithm). *Let $f \sim \mathcal{GP}(\mathbf{0}, k)$ with a stationary covariance function $k$. Assume that there exist constants $s_1, s_2 > 0$ such that $\mathbb{P}[sup_{\mathbf{x} \in \mathcal{X}} |\partial f / \partial x_i| > L] \leq s_1 e^{-(L/s_2)^2}$ for all $L > 0$ and for all $i \in \{1, 2, ..., d\}$. Pick a $\delta \in (0, 1)$. Thus, if $-1 \leq \alpha < 0$ then for any horizon $T > T_0$, the cumulative regret of the proposed HuBO algorithm is bounded as*

- $R_T \leq \mathcal{O}^*(T^{\frac{(\alpha+1)d+1}{2}})$ *if $k$ is a SE kernel,*

- $R_T \leq \mathcal{O}^*(T^{\frac{d2(\alpha+2)+d}{4\nu+2d(d+1)}})$ *if $k$ is a Matérn kernel*

*with probability greater than $1 - \delta$.*

*Proof.* By Proposition 1, we have $R_T \leq C + \sqrt{C_1 T \beta_T \gamma_T(\mathcal{C}_T)} + \frac{\pi^2}{6}$, where $\beta_T = 2log(4\pi_t/\delta) + 4dlog(dTs_2(b - a)(1 + \sum_{j=1}^T j^\alpha)\sqrt{log(4ds_1/\delta)})$. By Lemma 4, if $\alpha = -1$ then $\sum_{j=1}^T j^\alpha < 1 + ln(T)$, if $-1 < \alpha < 0$ then $\sum_{j=1}^T j^\alpha < 1 + \frac{T^{1+\alpha}-1}{1+\alpha}$. For both cases, $\beta_T \leq \mathcal{O}(log(T))$. By Proposition 2, the Theorem 2 holds. $\qquad\qquad\qquad\qquad\qquad\qquad\qquad\qquad\qquad\qquad\square$

## 4  Proof of Theorem 3

Figure 1: *An illustration of the case where a hypercube (the yellow square) intersects the sphere $S_\theta$ (the red circle) in two-dimensional space. In this case, the inscribed sphere (the yellow circle centered at $z_t^1$ with the radius $\frac{l_h}{2}$) of the hypercube intersects the sphere $S_\theta$ since $z_t^1$ is within the circle centered at $x^*$ with the radius $\theta + \frac{l_h}{2}$ (the green circle).*

**Theorem 3.** *Pick a $\delta \in (0, 1)$. Let $\mathbf{x}_t^* \in \mathcal{H}_t$ be the closest point to $\mathbf{x}^*$ in the search space $\mathcal{H}_t$. For any $t > T_0$ and $-1 \leq \alpha < 0$, with probability greater than $1 - \delta$, we have*

$$||\mathbf{x}_t^* - \mathbf{x}^*||_2 < \frac{2(b-a)}{\pi}(\Gamma(\frac{d}{2}+1))^{\frac{1}{d}}(log(\frac{1}{\delta}))^{\frac{1}{d}}M_t, \qquad (10)$$

*where the constant $T_0$ is defined in Theorem 1, $\Gamma$ is the gamma function, and $M_t = (2 + ln(t))t^{-\frac{\lambda}{d}}$ if $\alpha = -1$, otherwise, $M_t = 2(\alpha + 1)^{-1}t^{-\frac{\lambda}{d}}$ if $-1 < \alpha < 0$.*

*Proof.* The proof idea is to estimate the probability that $x_t^*$ lies in a sphere around $x^*$ with a small radius. Formally, we seek to bound $\mathbb{P}[||x_t^* - x^*||_2 \leq \theta]$ given a small $\theta > 0$.

It is hard to estimate directly $\mathbb{P}[||x_t^* - x^*||_2 \leq \theta]$. Instead, our idea is as follows. Since $x_t^* \in \mathcal{H}_t$, there exists a hypercube which contains $x_t^*$. We estimate the probability that this hypercube intersects the sphere $S_\theta = \{x \in \mathbb{R}^d | ||x - x^*||_2 \leq \theta\}$ which is centered at the optimum $x^*$ with the radius $\theta$.

We consider the case of $t > T_0$. By Theorem 1, $\mathcal{X}_t$ contains $x^*$ for every $t > T_0$, where $\mathcal{X}_t$ is the search space of the HuBO algorithm . We recall that $\mathcal{X}_t$ is different from $\mathcal{H}_t$ which is the search

space of the HD-HuBO algorithm that we are considering in this section. However, since the search space $\mathcal{H}_t$ is defined via $\mathcal{X}_t$, we need to use $\mathcal{X}_t$ to bound $\mathcal{H}_t$.

There are two cases to consider: **Case 1**: the whole sphere $S_\theta$ is within $\mathcal{X}_t$; **Case 2**: the only part of $S_\theta$ is within $\mathcal{X}_t$. Note that it is impossible that the whole sphere $S_\theta$ is outside of $\mathcal{X}_t$ since at least we have $x^* \in \mathcal{X}_t$ for $t > T_0$.

Figure 2: *An illustration of the case where the global optimum $x^*$ is a vertex of the square $\mathcal{X}_t$ in two-dimensional space. In this case, only a 1/4 volume of the sphere $S_{\theta+\frac{l_h}{2}}$ centered at $x^*$ with the radius $\theta + \frac{l_h}{2}$ (the green circle) is inside of $\mathcal{X}_t$.*

- **Case 1** where the whole sphere $S_\theta$ is within $\mathcal{X}_t$. We seek to bound the probability that a hypercube $H(z_t^i, l_h)$ intersects the sphere $S_\theta$, $1 \le i \le N_t$. We denote this probability by $p_0$. This probability is greater than the probability that the inscribed sphere of the hypercube $H(z_t^i, l_h)$, denoted by $S(z_t^i, \frac{l_h}{2})$ that has the center at $z_t^i$ and the radius $\frac{l_h}{2}$ intersects the sphere $S_\theta$. Let us define this probability as $p_1$. Further, $p_1$ is greater than the probability that the point $z_t^i$ is within the sphere around $x^*$ with the radius $\theta + \frac{l_h}{2}$. Let us define this probability as $p_2$. To explain the connection $p_1 \ge p_2$, we can see that the condition so that the sphere $S(z_t^i, \frac{l_h}{2})$ intersects the sphere $S_\theta$ is the distance between two centers $x^*$ and $z_t^i$ is less than or equal to the total of two radius. Figure 1 illustrates our situation.

  The probability $p_2$ can be computed by

  $$\frac{Vol(S_{\theta+\frac{l_h}{2}})}{Vol(\mathcal{X}_t)},$$

  where $Vol(\mathcal{X}_t)$ denotes the volume of the $\mathcal{X}_t$ and $Vol(S_{\theta+\frac{l_h}{2}})$ denotes the volume of the sphere $S_{\theta+\frac{l_h}{2}}$ centered at $x^*$ with the radius $\theta + \frac{l_h}{2}$.

  Since $p_0 > p_2$, we achieve

  $$p_0 > \frac{Vol(S_{\theta+\frac{l_h}{2}})}{Vol(\mathcal{X}_t)}.$$

  By Lemma 1, the volume of $\mathcal{X}_t$ can be computed as $Vol(\mathcal{X}_t) = ((b-a)(1 + \sum_{j=1}^t j^\alpha))^d$. By [5], the volume of the $d$-dimensional sphere with radius $\theta + \frac{l_h}{2}$ in $L^2$ norms is $\frac{(\pi(\theta+\frac{l_h}{2}))^d}{\Gamma(\frac{d}{2}+1)}$. Thus, the probability can be re-write as

  $$\frac{1}{\Gamma(\frac{d}{2}+1)}\Big[\frac{2\Gamma(\frac{3}{2})(\theta+\frac{l_h}{2})}{(b-a)(1+\sum_{j=1}^t j^\alpha))}\Big]^d.$$

- **Case 2** where the only part of $S_\theta$ is within $\mathcal{X}_t$. We only consider the case where $\theta+\frac{l_h}{2} < b-a$. Note that $b-a$ is the size of the initial space $\mathcal{X}_0$ as defined in Algorithm 1. $l_h$ denotes the size of hypercubes and $l_h$ is a parameter of the HD-HuBO algorithm. Hence, we can choose $l_h$ so that $l_h < b-a$ and $\theta < b-a-\frac{l_h}{2}$. It means that the sphere $S_\theta$ is small compared to $\mathcal{X}_t$.

  In the worst case where $x^*$ is at the boundary of $\mathcal{X}_t$ for all dimensions. See Figure 2 for an explanation. In this case, the size of the space part of $S_{\theta+\frac{l_h}{2}}$ in $\mathcal{X}_t$ halves in each

dimension and therefore, the volume of the space part of $S_{\theta + \frac{l_h}{2}}$ in $\mathcal{X}_t$, represented by $Vol(S_{\theta + \frac{l_h}{2}}) \cap Vol(\mathcal{X}_t)$ is reduced by $2^d$ times, compared to the whole volume of the sphere $S_{\theta + \frac{l_h}{2}}$. Thus, similar to Case 1, the probability $p_0$ that a hypercube $H(z_t^i, l_h)$ intersects the sphere $S_\theta$ is bounded as

$$p_0 > \frac{Vol(S_{\theta + \frac{l_h}{2}}) \cap Vol(\mathcal{X}_t)}{Vol(\mathcal{X}_t)} = \frac{1}{2^d} \frac{1}{\Gamma(\frac{d}{2}+1)} \left[ \frac{2\Gamma(\frac{3}{2})(\theta + \frac{l_h}{2})}{(b-a)(1 + \sum_{j=1}^t j^\alpha)} \right]^d.$$

Thus, in both Case 1 and Case 2, we have that the probability that a hypercube $H(z_t^i, l_h)$ intersects the sphere $S_\theta$ is bounded as

$$p_0 > \frac{1}{2^d} \frac{1}{\Gamma(\frac{d}{2}+1)} \left[ \frac{2\Gamma(\frac{3}{2})(\theta + \frac{l_h}{2})}{(b-a)(1 + \sum_{j=1}^t j^\alpha)} \right]^d.$$

It implies that the probability that a hypercube $H(z_t^i, l_h)$ does not intersect the sphere $S_\theta$ is computed as

$$
\begin{aligned}
1 - p_0 \quad &< \quad 1 - \frac{1}{2^d} \frac{1}{\Gamma(\frac{d}{2}+1)} \left[ \frac{2\Gamma(\frac{3}{2})(\theta + \frac{l_h}{2})}{(b-a)(1 + \sum_{j=1}^t j^\alpha)} \right]^d \\
&= \quad 1 - \frac{1}{\Gamma(\frac{d}{2}+1)} \left[ \frac{\Gamma(\frac{3}{2})(\theta + \frac{l_h}{2})}{(b-a)(1 + \sum_{j=1}^t j^\alpha)} \right]^d \\
&< \quad e^{-\frac{1}{\Gamma(\frac{d}{2}+1)} \left[ \frac{\Gamma(\frac{3}{2})(\theta + \frac{l_h}{2})}{(b-a)(1 + \sum_{j=1}^t j^\alpha)} \right]^d},
\end{aligned}
$$

where we use the inequality $1 - x \le e^{-x}$.

Therefore, if we consider the set of $N_t$ hypercubes then the probability that no hypercube $H(z_t^i, l_h)$ intersects the sphere $S_\theta$ is less than

$$\prod_{1 \le i \le N_t} e^{-\frac{1}{\Gamma(\frac{d}{2}+1)} \left[ \frac{\Gamma(\frac{3}{2})(\theta + \frac{l_h}{2})}{(b-a)(1 + \sum_{j=1}^t j^\alpha)} \right]^d} = e^{-N_t \frac{1}{\Gamma(\frac{d}{2}+1)} \left[ \frac{\Gamma(\frac{3}{2})(\theta + \frac{l_h}{2})}{(b-a)(1 + \sum_{j=1}^t j^\alpha)} \right]^d}$$

Note that this is achieved because the set of centres of hypercubes is sampled uniformly at random (hence independently). Thus, the probability that there is at least a hypercube from the set of $N_t$ hypercubes which intersects the sphere $S_\theta$ is at least:

$$1 - e^{-N_t \frac{1}{\Gamma(\frac{d}{2}+1)} \left[ \frac{\Gamma(\frac{3}{2})(\theta + \frac{l_h}{2})}{(b-a)(1 + \sum_{j=1}^t j^\alpha)} \right]^d}.$$

Further, since $l_h \ge 0$, $1 - e^{-N_t \frac{1}{\Gamma(\frac{d}{2}+1)} \left[ \frac{\Gamma(\frac{3}{2})(\theta + \frac{l_h}{2})}{(b-a)(1 + \sum_{j=1}^t j^\alpha)} \right]^d} \ge 1 - e^{-N_t \frac{1}{\Gamma(\frac{d}{2}+1)} \left[ \frac{\Gamma(\frac{3}{2})(\theta)}{(b-a)(1 + \sum_{j=1}^t j^\alpha)} \right]^d}$. Thus, the probability that there is at least a hypercube from the set of $N_t$ hypercubes which intersects the sphere $S_\theta$ is greater than:

$$1 - e^{-N_t \frac{1}{\Gamma(\frac{d}{2}+1)} \left[ \frac{\Gamma(\frac{3}{2})(\theta)}{(b-a)(1 + \sum_{j=1}^t j^\alpha)} \right]^d}.$$

Note that here, we omit the influence of the size of hypercubes. In fact, the larger the $l_h$, the higher the probability that there is at least a hypercube from the set of $N_t$ hypercubes which intersects the sphere $S_\theta$.

On the other hand, if let $x_t^* \in \mathcal{H}_t$ be the closest point to $x^*$ in the search space $\mathcal{H}_t$ then the probability that there is at least a hypercube from the set of $N_t$ hypercubes which intersects the sphere $S_\theta$ is equal to the probability that $\|x_t^* - x^*\|_2 \le \theta$. Thus, we have

$$\mathbb{P}[\|x_t^* - x^*\|_2 \le \theta] > 1 - e^{-N_t \frac{1}{\Gamma(\frac{d}{2}+1)} \left[ \frac{\Gamma(\frac{3}{2})(\theta)}{(b-a)(1 + \sum_{j=1}^t j^\alpha)} \right]^d}.$$

Now set $e^{-N_t \frac{1}{\Gamma(\frac{d}{2}+1)} \left[ \frac{\Gamma(\frac{3}{2})(\theta)}{(b-a)(1 + \sum_{j=1}^t j^\alpha)} \right]^d} = \delta$. We achieve $\theta = \frac{(b-a)}{\Gamma(\frac{3}{2})}(1 + \sum_{j=1}^t j^\alpha)(\Gamma(\frac{d}{2}+1))^{\frac{1}{d}} (\frac{1}{N_t} log(\frac{1}{\delta}))^{\frac{1}{d}} = \frac{2(b-a)}{\sqrt{\pi}}(1 + \sum_{j=1}^t j^\alpha)(\Gamma(\frac{d}{2}+1))^{\frac{1}{d}} (\frac{1}{N_t} log(\frac{1}{\delta}))^{\frac{1}{d}}$. Here, we use $\Gamma(\frac{3}{2}) = \frac{\sqrt{\pi}}{2}$.

Thus, given a $\delta \in (0, 1)$, we have

$$||x_t^* - x^*||_2 < \frac{2(b-a)}{\sqrt{\pi}}(1 + \sum_{j=1}^{t} j^\alpha)(\Gamma_t(\frac{d}{2} + 1))^{\frac{1}{d}}(\frac{1}{N_t}log(\frac{1}{\delta}))^{\frac{1}{d}},$$

with the probability $1 - \delta$.

By definition, $N_t = N_0 \lceil t^\lambda \rceil \geq t^\lambda$. Using the results from Lemma 2, we consider two cases of $\alpha$:

- if $\alpha = -1$, $1 + \sum_{j=1}^{t} \frac{1}{j} < 2 + ln(t)$. In this case, $||x_t^* - x^*||_2 \leq \frac{2(b-a)}{\sqrt{\pi}}(2 + ln(t))(\Gamma(\frac{d}{2} + 1))^{\frac{1}{d}}(\frac{1}{N_t}log(\frac{1}{\delta}))^{\frac{1}{d}} \leq \frac{2(b-a)}{\sqrt{\pi}}(\Gamma(\frac{d}{2} + 1))^{\frac{1}{d}}(log(\frac{1}{\delta}))^{\frac{1}{d}}\frac{2+ln(t)}{t^{\frac{\lambda}{d}}}$.

- if $-1 < \alpha < 0$, $1 + \sum_{j=1}^{t} j^\alpha < 2 + \frac{t^{\alpha+1}-1}{\alpha+1} = \frac{t^{\alpha+1}+2\alpha+1}{1+\alpha}$. Since $\alpha \leq 0$, $\frac{t^{\alpha+1}+2\alpha+1}{1+\alpha} \leq \frac{t^{\alpha+1}+1}{1+\alpha} \leq \frac{2}{\alpha+1}$. Thus, $||x_t^* - x^*||_2 \leq \frac{2(b-a)}{\sqrt{\pi}}(\Gamma(\frac{d}{2} + 1))^{\frac{1}{d}}(log(\frac{1}{\delta}))^{\frac{1}{d}}\frac{2}{\alpha+1}t^{-\frac{\lambda}{d}}$.

Let

$$M_t = \begin{cases} (2 + ln(t))t^{-\frac{\lambda}{d}}, & \text{if } \alpha = -1. \\ \frac{2}{\alpha+1}t^{-\frac{\lambda}{d}}, & \text{if } -1 < \alpha < 0. \end{cases}$$

, we have

$$||x_t^* - x^*||_2 < \frac{2(b-a)}{\sqrt{\pi}}(\Gamma(\frac{d}{2} + 1))^{\frac{1}{d}}(log(\frac{1}{\delta}))^{\frac{1}{d}}M_t,$$

with the high probability $1 - \delta$. The Theorem holds. $\qquad\square$

## 5  Proof of Theorem 4

Similar to HuBO, to derive the upper bounds of the cumulative regret of HD-HuBO for SE kernels and Matérn kernels, we first derive an upper bound of the cumulative regret for a general class of kernels as the following Proposition 3. We use Theorem 3 to prove this. Next, by combining results from Proposition 2 and Proposition 3, we achieve upper bounds for HD-HuBO for SE kernels and Matérn kernels.

**Proposition 3.** Let $f \sim \mathcal{GP}(\mathbf{0}, k)$ with a stationary covariance function $k$. Assume that there exist constants $s_1, s_2 > 0$ such that $\mathbb{P}[sup_{\mathbf{x}\in\mathcal{X}}|\partial f/\partial x_i| > L] \leq s_1 e^{-(L/s_2)^2}$ for all $L > 0$ and for all $i \in \{1, 2, ..., d\}$. Pick a $\delta \in (0, 1)$. Set $\beta_t = 2log(\pi^2 t^2/\delta) + 2dlog(2s_2 l_h d\sqrt{log(6ds_1/\delta)}t^2)$. Then, there exists a constant $C'$ such that with any horizon $T > T_0$, under conditions $\frac{\lambda}{d} > \alpha + 1$, $-1 \leq \alpha < 0$, $l_h > 0$, the cumulative regret of HD-HuBO Algorithm is bounded with probability greater than $1 - \delta$ as

$$R_T \leq C' + \sqrt{C_1 T \beta_T \gamma_T(\mathcal{C}_T)} + A(log(\frac{6}{\delta}))^{\frac{1}{d}}B_T + \frac{\pi^2}{6}, \text{ where } A = s_2\sqrt{log(\frac{l_h ds_1}{\delta})}\frac{2(b-a)}{\pi}d\sqrt{d+2},$$

and where $B_T = U_T V_T$ such that $U_T = 2 + ln(T)$ if $\alpha = -1$, otherwise $U_T = 2(\alpha + 1)^{-1}$, and $V_T = 1 + ln(T)$ if $\lambda = d$, otherwise $V_T = 1 + \frac{d}{d-\lambda}max\{1, T^{1-\frac{\lambda}{d}}\}$,

$C_1 = 8/log(1 + \sigma^2)$, $l_h$ is the size of the hypercube, $\mathcal{C}_T = [c_{min} - (b-a)(1 + \sum_{j=1}^{T} j^\alpha)/2, c_{max} + (b-a)(1 + \sum_{j=1}^{T} j^\alpha)/2]^d$, and $\gamma_T(\mathcal{C}_T)$ is the maximum information gain about the function $f$ from any $T$ observations from $\mathcal{C}_T$.

**Our Idea**  To derive a cumulative regret $R_T = \sum_{t=1}^{T} r_t$, we will seek to bound $r_t = f(x^*) - f(x_t)$ for any $t$. If $t \leq T_0$, similar to the proof of Proposition 2, we achieve a bound on $r_t$: $r_t \leq 2\sqrt{\beta_t}\sigma_{t-1}(x_t) + g'_t$, where $\beta_t$ is defined as in section 5 in the main paper, $g'_t$ is the is the gap between the global optimum and the optimum in $\mathcal{H}_t$. Formally, $g'_t = f(x^*) - f^*(\mathcal{H}_t)$.

Now we consider the case where $t > T_0$. Let $x_t^* \in \mathcal{H}_t$ be the closest point to $x^*$ in the search space $\mathcal{H}_t$. To obtain a bound on $r_t$ $(t > T_0)$, we write it as

$$\begin{align} r_t &= f(x^*) - f(x_t) \tag{11} \\ &= \underbrace{f(x^*) - f(x_t^*)}_{\text{Part 1}} + \underbrace{f(x_t^*)}_{\text{Part 2}} - \underbrace{f(x_t)}_{\text{Part 3}} \tag{12} \end{align}$$

Now we start to bound the part 1, the part 2 and part 3.

**Bounding Part 1**

**Lemma 7.** Pick a $\delta \in (0, 1)$. For any $t > T_0$, with probability at least $1 - \delta$, we have

$$|f(x^*) - f(x)| \leq s_2 \sqrt{log(\frac{2ds_1}{\delta})} \frac{2(b-a)}{\sqrt{\pi}} d\sqrt{d+2}(log(\frac{2}{\delta}))^{\frac{1}{d}} M_t,$$

where

$$M_t = \begin{cases} (2 + ln(t))t^{-\frac{\lambda}{d}}, & \text{if } \alpha = -1. \\ \frac{2}{\alpha+1}t^{-\frac{\lambda}{d}}, & \text{if } -1 < \alpha < 0. \end{cases}$$

*Proof.* Given any $x \in \mathcal{X}_t$, by Assumption of Theorem 4 and the union bound, we have,

$$|f(x^*) - f(x)| \leq L||x^* - x||_1$$

with probability greater than $1 - ds_1 e^{-L^2/s_2^2}$. Set $ds_1 e^{-L^2/s_2^2} = \delta/2$. Thus,

$$|f(x^*) - f(x)| \leq s_2 \sqrt{log(\frac{2ds_1}{\delta})}||x^* - x||_1 \tag{13}$$

with probability greater than $1 - \delta/2$.

On the other hand, By Theorem 3 we have:

$$||x_t^* - x^*||_2 \leq \frac{2(b-a)}{\sqrt{\pi}}(\Gamma(\frac{d}{2}+1))^{\frac{1}{d}}(log(\frac{2}{\delta}))^{\frac{1}{d}}M_t \tag{14}$$

with probability $1 - \delta/2$,

$$M_t = \begin{cases} (2 + ln(t))t^{-\frac{\lambda}{d}}, & \text{if } \alpha = -1. \\ \frac{2}{\alpha+1}t^{-\frac{\lambda}{d}}, & \text{if } -1 < \alpha < 0. \end{cases}$$

To transform from the $L^2$ norms to the $L^1$ norms, we use Cauchy-Schwarz:

$$||x_t^* - x^*||_1 \leq d||x_t^* - x^*||_2 \tag{15}$$

Combining Eq(13), Eq(14) and Eq(15), we have

$$|f(x^*) - f(x)| \leq s_2 \sqrt{log(\frac{2ds_1}{\delta})} d\frac{2(b-a)}{\sqrt{\pi}}(\Gamma(\frac{d}{2}+1))^{\frac{1}{d}}(log(\frac{2}{\delta}))^{\frac{1}{d}}M_t$$

with the probability $1 - \delta$.

Further, by Lemma 6, we achieve $(\Gamma(\frac{d}{2}+1))^{\frac{1}{d}} < \sqrt{d+2}$. Thus,

$$|f(x^*) - f(x)| \leq s_2 \sqrt{log(\frac{2ds_1}{\delta})} \frac{2(b-a)}{\sqrt{\pi}} d\sqrt{d+2}(log(\frac{2}{\delta}))^{\frac{1}{d}}M_t$$

with the probability $1 - \delta$. $\qquad \square$

**Bounding Part 2** Now, we continue to bound the part 2. By definition, $x_t^* \in \mathcal{H}_t$. Since $\mathcal{H}_t = \{H(\mathbf{z}_t^1, l_h) \cup ... \cup H(\mathbf{z}_t^{N_t}, l_h)\} \cap \mathcal{X}_t$, $x_t^*$ is in some hypercube. Without the loss of generality, we assume that $x_t^*$ is within the hypercube $H(z_t^*, l_h)$, where $z_t^*$ is one centre among sampled centres $\{z_t^1, ..., z_t^{N_t}\}$.

**Lemma 8** (Bounding Part 2)**.** Pick a $\delta \in (0, 1)$ and set $\zeta_t^1 = 2log(\frac{\pi^2 t^2}{3\delta}) + 2dlog(2s_1 l_h d\sqrt{log(\frac{2ds_1}{\delta})}t^2)$. Then, there exists a $x' \in H(z_t^*, l_h)$ such that

$$f(x_t^*) \leq \mu_{t-1}(x') + \sqrt{\zeta_t^1}\sigma_{t-1}(x') + \frac{1}{t^2} \tag{16}$$

holds with probability $\geq 1 - \delta$.

*Proof.* We use the idea of proof of Lemma 5.7 in [4] for the hypercube $H(z_t^*, l_h)$. We consider the distance of any two points in the hypercube: $||x - x'||_1$. We have $||x - x'||_1 \leq l_h$, where $l_h$ is the size of the hypercube.

By Assumption of Theorem 4 and the union bound, for $\forall x, x'$, we have

$$|f(x) - f(x')| \leq L||x - x'||_1$$

with probability greater than $1 - ds_1 e^{-L^2/s_2^2}$. Thus, by choosing $ds_1 e^{-L^2/s_2^2} = \delta/2$, we have

$$|f(x) - f(x')| \leq s_2 \sqrt{log(\frac{2ds_1}{\delta})}||x - x'||_1 \tag{17}$$

with probability greater than $1 - \delta/2$.

Now, on $H(z_t^*, l_h)$, we construct a discretization $F_t$ of size $(\tau_t)^d$ dense enough such that for any $x \in F_t$

$$||x - [x]_t||_1 \leq \frac{l_h d}{\tau_t}$$

where $[x]_t$ denotes the closest point in $F_t$ to $x$. In this manner, with probability greater than $1 - \delta/2$, we have

$$
\begin{aligned}
|f(x) - f([x]_t)| &\leq & \frac{s_2 \sqrt{log(\frac{2ds_1}{\delta})}||x - [x]_t||_1}{\tau_t} \\
&\leq & s_2 \sqrt{log(\frac{2ds_1}{\delta})} \frac{l_h d}{\tau_t} \\
&< & \frac{s_2 l_h d \sqrt{log(\frac{2ds_1}{\delta})}}{\tau_t}
\end{aligned}
$$

Here, we use the inequality $||x - [x]_t||_1 \leq l_h d$. Let $\tau_t = s_2 l_h d \sqrt{log(\frac{2ds_1}{\delta})}t^2$. Thus, $|F_t| = (s_2 l_h d \sqrt{log(\frac{2ds_1}{\delta})}t^2)^d$. We obtain

$$|f(x) - f([x]_t)| \leq \frac{1}{t^2} \tag{18}$$

with probability $1 - \delta/2$ for any $x \in F_t$.

Similar to Lemma 5.6 of [4], if we set $\zeta_t^1 = 2log(|F_t|\frac{\pi^2 t^2}{3\delta}) = 2log(\frac{\pi^2 t^2}{3\delta}) + 2dlog(s_2 l_h d \sqrt{log(\frac{2ds_1}{\delta})}t^2)$, we have with probability $1 - \delta/2$, we have

$$f(x) \leq \mu_{t-1}(x) + \sqrt{\zeta_t^1}\sigma_{t-1}(x) \tag{19}$$

for any $x \in F_t$ and any $t \geq 1$. Thus, combining Eq(18) and Eq(19), if we let $[x]_t$ which is the closest point in $F_t$ to $x$, we have

$$f(x_t^*) \leq \mu_{t-1}([x_t^*]_t) + \sqrt{\zeta_t^1}\sigma_{t-1}([x_t^*]_t) + \frac{1}{t^2}$$

with probability $1 - \delta$. $\qquad\square$

**Bounding Part 3**

**Lemma 9.** Pick a $\delta \in (0, 1)$ and set $\zeta_t^0 = 2log(\pi^2 t^2/(6\delta))$. Then we have

$$f(x_t) \geq \mu_{t-1}(x_t) - \sqrt{\zeta_t^0}\sigma_{t-1}(x_t) \tag{20}$$

holds with probability $\geq 1 - \delta$.

*Proof.* It is similar to Lemma 5.5 of [4]. $\qquad\square$

Now, we combine the results from Lemmas 7, 8 and 9 to obtain a bound on $r_t$ as in the following Lemma.

**Lemma 10** (Bounding $r_t$). Pick a $\delta \in (0, 1)$ and set $\beta_t = 2log(\frac{\pi^2 t^2}{\delta}) + 2dlog(2s_2 l_h d \sqrt{log(\frac{6s_1 d}{\delta})}t^2)$. Then with $t > T_0$, $\frac{\lambda}{d} > \alpha + 1$, $-1 \le \alpha < 0$ and $l_h > 0$, we have

$$r_t \le 2\beta_t^{1/2}\sigma_{t-1}(x_t) + \frac{1}{t^2} + A(log(\frac{6}{\delta}))^{\frac{1}{d}} M_t \qquad (21)$$

holds with probability $\ge 1 - \delta$, where $A = s_2 \sqrt{log(\frac{2ds_1}{\delta})}\frac{2(b-a)}{\sqrt{\pi}}d\sqrt{d+2}$ and

$$M_t = \begin{cases} (2 + ln(t))t^{-\frac{\lambda}{d}}, & \text{if } \alpha = -1. \\ \frac{2}{\alpha+1}t^{-\frac{\lambda}{d}}, & \text{if } -1 < \alpha < 0. \end{cases}$$

*Proof.* We use $\frac{\delta}{3}$ for Lemmas 7, 8 and 9 so that these events hold simultaneously with probability greater than $1 - \delta$. Formally, by Lemma 9 using $\frac{\delta}{3}$:

$$f(x_t) \ge \mu_{t-1}(x_t) - \sqrt{2log(\frac{\pi^2 t^2}{\delta})}\sigma_{t-1}(x_t)$$

holds with probability $\ge 1 - \frac{\delta}{3}$. As a result,

$$
\begin{align}
f(x_t) &\ge \mu_{t-1}(x_t) - \sqrt{\zeta_t^0}\sigma_{t-1}(x_t) \qquad (22) \\
&> \mu_{t-1}(x_t) - \sqrt{\beta_t}\sigma_{t-1}(x_t) \qquad (23)
\end{align}
$$

holds with probability $\ge 1 - \frac{\delta}{3}$.

By Lemma 8 using $\frac{\delta}{3}$, there exists a $x' \in H(z_t^*, l_h)$ such that

$$f(x_t^*) \le \mu_{t-1}(x') + \sqrt{\zeta_t^1}\sigma_{t-1}(x') + \frac{1}{t^2}$$

holds with probability $\ge 1 - \frac{\delta}{3}$. As a result,

$$
\begin{align}
f(x_t^*) &\le \mu_{t-1}(x') + \sqrt{\zeta_t^1}\sigma_{t-1}(x') + \frac{1}{t^2} \\
f(x_t^*) &\le \mu_{t-1}(x') + \sqrt{\beta_t}\sigma_{t-1}(x') + \frac{1}{t^2} \\
&= u_t(x') + \frac{1}{t^2}
\end{align}
$$

Recall that $u_t(x)$ is the acquisition function defined in the main paper. Since $x_t = \text{argmax}_{x \in \mathcal{X}'_t} u_t(x)$ and $x' \in H(z_t^*, l_h) \subset \mathcal{X}'_t$, we have $u_t(x') \le u_t(x_t)$. Thus,

$$f(x_t^*) \le u_t(x_t) + \frac{1}{t^2} \qquad (24)$$

holds with probability $\ge 1 - \frac{\delta}{3}$.

By Lemma 7 using $\frac{\delta}{3}$:

$$|f(x_t^*) - f(x^*)| \le A(log(\frac{6}{\delta}))^{\frac{1}{d}} M_t \qquad (25)$$

holds with probability $\ge 1 - \frac{\delta}{3}$.

Combing Eq(23), Eq(24) and Eq(25), we have

$$
\begin{align}
r_t &= f(x^*) - f(x_t) \tag{26}\\
&= \underbrace{f(x^*) - f(x_t^*)}_{\text{Part 1}} + \underbrace{f(x_t^*)}_{\text{Part 2}} - \underbrace{f(x_t)}_{\text{Part 3}} \tag{27}\\
&\leq A(log(\frac{6}{\delta}))^{\frac{1}{d}}M_t + \underbrace{f(x_t^*)}_{\text{Part 2}} - \underbrace{f(x_t)}_{\text{Part 3}} \tag{28}\\
&\leq A(log(\frac{6}{\delta}))^{\frac{1}{d}}M_t + \frac{1}{t^2} + u_t(x_t) - f(x_t) \tag{29}\\
&\leq A(log(\frac{6}{\delta}))^{\frac{1}{d}}M_t + \frac{1}{t^2} + 2(\beta_t)^{1/2}\sigma_{t-1}(x_t) \tag{30}
\end{align}
$$

holds with probability $\geq 1 - \delta$. $\qquad\qquad\square$

Now we are ready to prove Proposition 3.

We have $R_T = \sum_{t=1}^{T} r_t = \sum_{t=1}^{T_0} r_t + \sum_{t=T_0+1}^{T} r_t$.

Similar to the proof of Proposition 1, we have $\sum_{t=1}^{T_0} r_t \leq \sum_{t=1}^{T_0}(2\sqrt{\beta_t}\sigma_{t-1}(x_t) + g_t')$.

On the other hand, By Lemma 10, we have $\sum_{t=T_0+1}^{T} r_t \leq \sum_{t=T_0+1}^{T}(2\beta_t^{1/2}\sigma_{t-1}(x_t) + \frac{1}{t^2} + A(log(\frac{6}{\delta}))^{\frac{1}{d}}M_t)$. Thus, $R_T = \sum_{t=1}^{T} r_t \leq \sum_{t=1}^{T_0} g_t' + \frac{\pi^2}{6} + \sum_{t=1}^{T} 2\beta_t^{1/2}\sigma_{t-1}(x_t) + \sum_{t=T_0+1}^{T} A(log(\frac{6}{\delta}))^{\frac{1}{d}}M_t$. We set $C' = \sum_{t=1}^{T_0} g_t'$. To make our problem in context of unknown search spaces tractable, we assume that the function $f$ is *finite* on any finite domain of $\mathbb{R}^d$. It implies that for every $1 \leq t \leq T_0$, $g_t'$ is finite. Further, by definition of $T_0$, $T_0$ is the constant and independent of $T$. Thus, $C'$ is also a constant and is independent of $T$. Thus, we have $R_T \leq C' + \frac{\pi^2}{6} + \sum_{t=1}^{T} 2\beta_t^{1/2}\sigma_{t-1}(x_t) + \sum_{t=T_0+1}^{T} A(log(\frac{6}{\delta}))^{\frac{1}{d}}M_t$

To bound $\sum_{1}^{T} 2\beta_t^{1/2}\sigma_{t-1}(x_t)$, we use the property of $\mathcal{C}_T$ and $\mathcal{H}_T$ that $\mathcal{H}_T \subseteq \mathcal{X}_T \subseteq \mathcal{C}_T$. Hence, similar to the proof of Lemma 5.4 of [4], we have $\sum_{1}^{T} 2\beta_t^{1/2}\sigma_{t-1}(x_t) \leq \sqrt{C_1 T \beta_T \gamma_T(\mathcal{C}_T)}$. The remaining problem is to bound $\sum_{T_0}^{T} A(log(\frac{6}{\delta}))^{\frac{1}{d}}M_t = A(log(\frac{6}{\delta}))^{\frac{1}{d}}\sum_{T_0}^{T} M_t$, where

$$
M_t = \begin{cases} (2 + ln(t))t^{-\frac{\lambda}{d}}, & \text{if } \alpha = -1. \\ \frac{2}{\alpha+1}t^{-\frac{\lambda}{d}}, & \text{if } -1 < \alpha < 0. \end{cases}
$$

We consider two cases of $\alpha$:

- If $\alpha = -1$, then $\sum_{t=T_0}^{T} M_t \leq \sum_{t=1}^{T} \frac{2 + ln(t)}{t^{\frac{\lambda}{d}}} < (2 + ln(T))\sum_{t=1}^{T} \frac{1}{t^{\frac{\lambda}{d}}}$. We consider three cases of $\lambda$:
  - if $\lambda = d$, $\sum_{t=1}^{T} \frac{1}{t^{\frac{\lambda}{d}}} = \sum_{t=1}^{T} \frac{1}{t} < 1 + ln(T)$ (using Lemma 4). Therefore, $\sum_{T_0}^{T} A(log(\frac{6}{\delta}))^{\frac{1}{d}}M_t < A(log(\frac{6}{\delta}))^{\frac{1}{d}}B_T$, where $B_T = (2 + ln(T))(1 + ln(T))$.
  - if $\lambda > d$, $\sum_{t=1}^{T} \frac{1}{t^{\frac{\lambda}{d}}} < 1 + \frac{1}{\lambda/d-1} = \frac{\lambda}{\lambda-d}$(using Lemma 5). Thus, $\sum_{T_0}^{T} A(log(\frac{6}{\delta}))^{\frac{1}{d}}M_t < A(log(\frac{6}{\delta}))^{\frac{1}{d}}B_T$, where $B_T = (2 + ln(T))(1 + \frac{d}{d-\lambda})$.
  - if $0 < \lambda < d$, $\sum_{t=1}^{T} \frac{1}{t^{\frac{\lambda}{d}}} < 1 + \frac{T^{1-\frac{\lambda}{d}}}{1-\frac{\lambda}{d}} < 1 + \frac{d}{d-\lambda}T^{1-\frac{\lambda}{d}}$. Thus, $\sum_{T_0}^{T} A(log(\frac{6}{\delta}))^{\frac{1}{d}}M_t < A(log(\frac{6}{\delta}))^{\frac{1}{d}}B_T$, where $B_T = (2 + ln(T))(1 + \frac{d}{d-\lambda}T^{1-\frac{\lambda}{d}})$.

- If $-1 < \alpha < 0$, then $\sum_{t=T_0}^{T} M_t \leq \frac{2}{\alpha+1}(\sum_{t=1}^{T} \frac{1}{t^{\frac{\lambda}{d}}})$. Similar to the above case, we consider three cases of $\lambda$:
  - if $\lambda = d$, $\sum_{t=1}^{T} \frac{1}{t^{\frac{\lambda}{d}}} = \sum_{t=1}^{T} \frac{1}{t} < 1 + ln(T)$ (using Lemma 4). Therefore, $\sum_{T_0}^{T} A(log(\frac{6}{\delta}))^{\frac{1}{d}}M_t < A(log(\frac{6}{\delta}))^{\frac{1}{d}}B_T$, where $B_T = \frac{2}{\alpha+1}(1 + ln(T))$.
  - if $\lambda > d$, $\sum_{t=1}^{T} \frac{1}{t^{\frac{\lambda}{d}}} < 1 + \frac{1}{\lambda/d-1} = \frac{\lambda}{\lambda-d}$(using Lemma 5). Thus, $\sum_{T_0}^{T} A(log(\frac{6}{\delta}))^{\frac{1}{d}}M_t < A(log(\frac{6}{\delta}))^{\frac{1}{d}}B_T$, where $B_T = \frac{2}{\alpha+1}(1 + \frac{d}{d-\lambda})$.

– if $0 < \lambda < d$, $\sum_{t=1}^{T} \frac{1}{t^{\frac{\lambda}{d}}} < 1 + \frac{T^{1-\frac{\lambda}{d}}}{1-\frac{\lambda}{d}} < 1 + \frac{d}{d-\lambda}T^{1-\frac{\lambda}{d}}$. Thus, $\sum_{T_0}^{T} A(log(\frac{6}{\delta}))^{\frac{1}{d}}M_t <$
$A(log(\frac{6}{\delta}))^{\frac{1}{d}}B_T$, where $B_T = \frac{2}{\alpha+1}(1 + \frac{d}{d-\lambda}T^{1-\frac{\lambda}{d}})$.

For all cases, with probability greater than $1 - \delta$ we achieve $R_T \leq C' + \sqrt{C_1 T \beta_T \gamma_T(\mathcal{C}_T)} + A(log(\frac{6}{\delta}))^{\frac{1}{d}}B_T + \frac{\pi^2}{6}$ , where $A = s_2\sqrt{log(\frac{l_h d s_1}{\delta})}\frac{2(b-a)}{\pi}d\sqrt{d+2}$, and $B_T = U_T V_T$ such that $U_T = 2 + ln(T)$ if $\alpha = -1$, otherwise $U_T = 2(\alpha+1)^{-1}$, and $V_T = 1 + ln(T)$ if $\lambda = d$, otherwise $V_T = 1 + \frac{d}{d-\lambda}\max\{1, T^{1-\frac{\lambda}{d}}\}$. Thus, Proposition 3 holds.

**Theorem 4** (Cumulative Regret $R_T$ of HD-HuBO Algorithm). Let $f \sim \mathcal{GP}(\mathbf{0}, k)$ with a stationary covariance function $k$. Assume that there exist constants $s_1, s_2 > 0$ such that $\mathbb{P}[sup_{\mathbf{x}\in\mathcal{X}}|\partial f/\partial x_i| > L] \leq s_1 e^{-(L/s_2)^2}$ for all $L > 0$ and for all $i \in \{1, 2, ..., d\}$. Pick a $\delta \in (0, 1)$. Then, with $T > T_0$, under conditions $\lambda > d(\alpha+1)$, $-1 \leq \alpha < 0$, $l_h > 0$, the cumulative regret of proposed HD-HuBO algorithm is bounded as

- $R_T \leq \mathcal{O}^*(T^{\frac{(\alpha+1)d+1}{2}} + (log(\frac{6}{\delta}))^{\frac{1}{d}}B_T)$ if $k$ is a SE kernel,

- $R_T \leq \mathcal{O}^*(T^{\frac{d^2(\alpha+2)+d}{4\nu+2d(d+1)}+\frac{1}{2}} + (log(\frac{6}{\delta}))^{\frac{1}{d}}B_T)$ if $k$ is a Matérn kernel,

with probability greater than $1 - \delta$, where $B_T = U_T V_T$ such that $U_T = 2 + ln(T)$ if $\alpha = -1$, otherwise $U_T = 2(\alpha+1)^{-1}$, and $V_T = 1 + ln(T)$ if $\lambda = d$, otherwise $V_T = 1 + \frac{d}{d-\lambda}\max\{1, T^{1-\frac{\lambda}{d}}\}$.

*Proof.* Theorem holds due to Proposition 2 and Proposition 3. □

# 6 Experiments

Figure 3: Comparison of baselines and the proposed methods when the initial search space is very small fraction ($2\%$) of the pre-defined space.

**On the initial search space**  The initial search space is crucial to the optimisation efficiency of any volume expansion strategy. However, since the search space is unknown, in reality it is possible that the initial search domain is very far from the global optimum. We consider this situation by setting the initial search space to be only $2\%$ of the pre-defined domain. Under this setting, we optimise two functions: Hartmann6 and 5-dims Ackley function. As seen in Figure 3, our algorithms outperform baselines due to the expansion and especially translations of search spaces toward the promising regions. This is a benefit of our algorithm compared to the previous works in unknown search spaces.

**On the computational effectiveness**  The computational time is an important benefit for our algorithms. In our experiments, $\mathcal{C}_{initial}$ is set to 10 times to the size of the initial search space $\mathcal{X}_0$ along each dimension, it allows expanded spaces to move freely to any position in the pre-defined domain. It follows that via the transformation, the center of the new search space is set closer to the best solution found up to that iteration. Therefore, both the new bound and the new center are easy to determine compared to previous works in unknown search spaces except the volume doubling strategy. We note that in practice, if the search domain is unknown, our algorithm would typically benefit by setting a large $\mathcal{C}_{initial}$ as this allows the search space to be centered close to the best found solution.

Figure 4: The average runtime (seconds) of HD-HuBO over iterations.

Table 1: Average CPU time (seconds) at the final iteration for all algorithms.

| Algorithms | Beale | Hartmann3 | Hartmann6 | Levy(d =20) | Ackley(d =20) |
|---|---|---|---|---|---|
| HuBO | 0.40 | 0.74 | 3.06 | 6.63 | 9.98 |
| HD-HuBO | 0.48 | 0.76 | 3.14 | 6.90 | 11.13 |
| Re-H | 0.49 | 0.84 | 0.91 | 7.22 | 12.96 |
| Re-Q | 0.47 | 1.52 | 6.12 | 6.97 | 13.21 |
| Vol2 | 0.37 | 0.76 | 2.89 | 6.34 | 9.13 |
| UBO | 0.61 | 2.11 | 11.21 | 9.37 | 21.33 |
| FBO | 1.91 | 4.32 | 29.50 | 23.67 | 46.56 |

For HD-HuBO, to optimise over multiple disjoint hypercubes in the continuous input space, we perform optimisation for each hypercube and then take the best maximum value found across all hypercubes. For example, for synthetic functions we used $\lambda = 1, N_0 = 1$ and thus $N_t = t$. This means that at iteration $t$, we use $t$ hypercubes for the maximisation of acquisition function. We optimise the acquisition function using L-BFGS with 20 restarts on each hypercube. The maximum number of acquisition function evaluations is set to 1000. The Figure 4 shows the average runtime (seconds) of HD-HuBO over iterations on the 20-dims Levy function.

To compare the computational time of all algorithms, we give to all the algorithms the equal computational budget to maximise acquisition functions at each iteration. As seen in Table 1, our algorithms are faster than UBO which needs to compute singular values of matrix $(\mathbf{K} + \sigma\mathbf{I})^{-1}$, and faster than FBO, which needs extra steps to numerically solve multiple optimisation problems for FBO.

Figure 5: Optimisation efficiency with different sizes of the search space

**Additional Results**   When the search space is *unknown*, one heuristic solution is to specify it arbitrarily. However, there are two problems: (1) an arbitrary search space that is finite, no matter how large, may not contain the global optimum (2) optimisation efficiency decreases with increasing size of the search space. We below provide two examples to illustrate that the optimisation efficiency decreases with increasing size of the search space.

In low dimensions, we consider the optimisation efficiency of BO algorithms such as EI and GP-UCB on 5-dims Levy function when increasing the size of the search space. We consider two cases: (1) the search space is set to $[-10, 10]$ and (2) the search space is set to $[-100, 100]$. In high dimensions, we consider the optimisation efficiency of REMBO algorithm [6] and LINEBO algorithm [3] on 20-dims Levy function. Also, we consider two cases: (1) the search space is set to $[-10, 10]$ and (2) the search space is set to $[-100, 100]$. The Levy function achieves the minimum value at $x^* = (1, 1, ..., 1)$.

As seen in Figure 5, the use of a larger space slows down fast the convergence. In contrast, our approach using a volume expansion strategy starting from a small initial search space can avoid this unnecessary sampling.