[Reviews · NeurIPS 2020]

Review 1

Summary and Contributions: This paper proposes new algorithms for unconstrained bayesian optimization problem. The first algorithm, HuBO, expands the search space while translating the center of the search space. The second one, HD-HuBO, further restricts the search space within the union of smaller hypercubes. Both algorithms have theoretical convergence guarantees.

Strengths: HuBO is the first (according to the authors; I did not verify this claim) to achieve global convergence to the exact optimum on unknown search space. The convergence analysis technique is quite interesting and gave me a useful insight; HuBO works because the radius of search space expands logarithmically in t, and the maximum information gain doesn't grow faster than that. The algorithm can be useful in practice.

Weaknesses: I think that the most important applications of the proposed algorithm is when x* is very far from a human-developed search range X_0. It is OK that the theory treats dist(X_0, x*) as a constant in T, but the experimental results don't give enough evidence on the algorithm's applicability to problems where dist(X_0, x*) is large.

Correctness: I didn't verify the proof of their theorems (in appendix), but their mathematical claims and proof approach seem reasonable. For the experiments, I have a few questions. 1. In line 240, the authors said that the initial search space was randomly placed. The regret curves have a very tight confidence bound, starting from the very first iterations. Shouldn't it vary more? Also, did you use the same X_0 for all compared algorithms? 2. X_0 being 20% of [0,1]^d doesn't seem to put x* sufficiently farm from X_0. When x* is much farther from X_0, the logarithmic growth rate of the radius X_t would be too slow, compared to Vol2, for example. It'd be nice to see the HuBO performance as a function of distance between X_0 and x*.

Clarity: The paper is clearly written and it's easy to follow. One minor comment is that \lambda=1 and \alpha=-1 seem to be clear winners both in theory and practice. So perhaps you could simplify the main theorem statement after fixing \lambda and \alpha? The statement and proofs for other values of \lambda and \alpha can go into appendix.

Relation to Prior Work: The related work section does a good job putting this work into context.

Reproducibility: No

Additional Feedback: Algorithm 2. X_t is never defined. I assumed that X_t is defined by Equation 2 like Algorithm 1. Authors mentioned the same computational budget for acquisition function optimization. What is the optimizer though? Constrained optimization of the acquisition function inside H_t (Equation 3) does not seem trivial. It isn't mentioned anywhere how the acquisition funciton was optimized. I think results in Figure 2 warrant more explanations. For HD-HuBO, l_h is 10% of X_0. And t of them is used at iteration t. In the meantime, the radius of X_t increases in log(t). So I believe that HuBO and HD-HuBO should become identical (once H_t covers the full search space) at a relatively small t and such t is determined independently of the dimensionality. Why do we see more pronounced performance difference in high-dimensional problems? In addition, HD-HuBO has worse regret guarantee (line 215) but performs better in experiments. Is the performance gain is purely from acquisition funciton optimization? (This goes back to my previous question of what optimzier did you use?) How does the GP-UCB perform given the full knowledge of search space? It'd be especially interesting for high-d cases. Should Proposition 2 be stated in O* instead of O? For example, if alpha=-1, then for SE kernels, I get \gamma_T = O(T^0)=O(1), but in the proof it says \gamma_T <= O((log T)^d). I think this paper is solid overall, and contributes an interesting theoretical result. The experiments section could be improved.


Review 2

Summary and Contributions: The paper proposes two new algorithms for Bayesian optimization when the input domain is unknown. The hey aspect of the methods is a new way of expanding and adapting a sequence on input domains where UCB search is performed. Two variants of the algorithm are proposed. One in which the boundaries some initial box are alpha-expanded and the domain re-centered. A second one, more focused on hight dimensional problems in which boxes around collected points are are taken into account. Both algorithms are analysed theoretically, and an experimental evaluation show that they work better in practice than alternative methods.

Strengths: - The paper addresses is not yet well understood problem. - The algorithm is new and well presented. - The algorithm is analysed theoretically. The authors show that it achieves sub-linear regret.

Weaknesses: I question how much this method (and similar methods of this type) can be used in practice. In most problems we have an intuition of where the optimum sits, and that intuition is reflected in the way we construct the input domain. In a way, we can think about the initial input domain as some uniform prior of the location of the optimum. This methods also requires of such intuition because an initial input space is also required. Starting from there, the authors propose to expand the space. Wouldn't it be a better strategy to use our intuition to define a larger space and find strategies to reduce it?

Correctness: The paper is technically correct.

Clarity: Yes, the paper is well written and clear. It would benefit of some graphical explanations in some toy problems (2D) of how the algorithms work but this is not strictly necessary.

Relation to Prior Work: Prior work has been cited in general although some recent references to convergence rates in BO using UCB are missing. Bandit optimisation of functions in the Matern kernel RKHS. David Janz, David R Burt, Javier González. Artificial intelligence and Statistics, AISTATS 2020.

Reproducibility: Yes

Additional Feedback: I think that this is a new approach that has been correctly executed but, following my comment above, I still question that starting with a small domain and expand it is better than starting with large domain and try to reduce to areas where the optimum may belong to. To give a precise example. If I am tuning the regularisation parameter \lambda of a logistic regression such parameter will belong to [0, infty). Is it reasonable to think that lambda = 10^99999? Probably not, which means that I can always defined a compact subset where I should be doing the optimisation. If the authors convince me that this is not the case, I am willing to improve my score. ================================== I have read the rebuttal and I have changed my score. ==================================


Review 3

Summary and Contributions: In standard Bayesian optimization, the search space X_0 is fixed up front. If HPs are not well understood, overly wide ranges can mean slow convergence, and overly narrow ones risk missing the best solution. This work proposed HuBO, where the search space X_t is slowly expanded and recentered in each round. By bounding volumes of these search spaces, they obtain regret bounds, close to the fixed search space case for the slowest expansion. In high dimensions, the acqusition function (GP-UCB) is hard to globally optimize over a expanding space. In their HD-HuBO, they sample N_t points from X_t at random and only search in small (fixed) sized volumes around them. Regret bounds are obtained for HD-HuBO as well. While larger than for HuBO, they are still sublinear for slow HuBO expansion and modest growth of N_t (e.g., linear). In their experiments, their two methods are compared to a range of previous work for BO with unknown search spaces, on synthetic blackbox functions, and a few ML HPO problems (elastic net, RL, and training a small MLP with BO). I read the author response and see no reason to change my score.

Strengths: This work provides new BO algorithms which expands and moves its search space in each round. Its particular choice of expansion is theoretically grounded, pointing out it should be slow enough. I could not really follow the theory, but it does not contradict intuition. Next, and difficult to earlier theoretical work on GP-UCB, they point out that optimizing the acquisition function (AF) in high dimensions is hard in practice, rendering theoretical results intractable. Their HD-HuBO proposes a particular heuristic to optimize the AF, namely local search in fixed-size regions around randomly sampled points. If the number N_t of these seed points grows aligned with the expansion of X_t, sublinear regret is also obtained for this heuristic. Their theoretical results are practically relevant. Also, in their experiments, both proposed methods outperform baselines, and HD-HuBO outperforms HuBO for higher dimensions, indicating that the AF optimization in HuBO did not find global optima.

Weaknesses: This is a dense paper with a lot of theory. I'd like the authors to provide more intuition in some cases. For example, what is the drawback of always just setting alpha = -1? It seems that T_0 is larger, but by how much? I also find it odd that Theo 3 and 4 do not depend on l_h, can I let l_h -> 0? I also do not understand how the theoretical results depend on the size of the initial space X_0. Surely, if I make this very small, things can go wrong? I feel that both the size of X_0 and alpha -> -1 must prominently feature in beta_t, which has to grow fast enough such that the method does not converge prematurely before any evaluations can be close to x_*. The experimental results do not have sufficient detail in order to reproduce results. They compare against a range of prior ideas to work with an unspecified search space. However, I am missing obvious baselines: - Standard GP-UCB with wide search ranges - Standard GP-UCB with wide search ranges plus their HD heuristic of sampling N_t points to start from I also do not understand the rationale for learning neural network weights with HPO, why not use gradient descent for that?

Correctness: The proofs in the supplemental material are beyond my ability to check. Except for what I said in "weaknesses", the theory does not contradict my intuition. Since their result is an extension of earlier work on GP-UCB with fixed search space, it would be good if they motivated what is different here, and how beta_t has to be chosen differently.

Clarity: This is a dense paper with a number of theoretical results. The algorithms are clear, as is the motivation for them. I feel the theoretical results could be better related to earlier ones with fixed space, pointing out the differences, and also the core ideas that make the proofs work. As it is, results have to be taken on faith, unless a very dense supplement is worked through, where previous techniques are used in a somewhat off-hand way.

Relation to Prior Work: There is a good list of related work.

Reproducibility: No

Additional Feedback:


Review 4

Summary and Contributions: This work proposes an unknown Bayesian optimization method to optimize an unknown black-box function on an unknown search space. Since, in practice, we do not know a feasible bound of Bayesian optimization, this problem is interesting. This paper solves it expanding a search space with the rate of expansion through a hyperharmonic series. The authors provide the theoretical and empirical results.

Strengths: + Motivation is strong. + Related work is well-organized. + Theoretical results are interesting.

Weaknesses: - Experimental results are somewhat strange to me. - Theoretical results have a limitation where small $\alpha$ is given. Please check the description below.

Correctness: It seems correct, although I didn't check all the proofs. However, I have a concern on small $\alpha$.

Clarity: This paper is well-written and well-organized.

Relation to Prior Work: There are some works on unbounded Bayesian optimization, but this work carefully describe and compare the difference to them.

Reproducibility: No

Additional Feedback: As described above, the topic handled in this paper is very interesting and important. However, I have two concerns on the theoretical and empirical results. I think small $\alpha$ cannot be avoidable. For example, if $-1 + 1 /d \leq \alpha \leq 0$ for SE kernels, a sublinear regret cannot be satisfied. It is pretty critical, since if $d = 2$ $-0.5 \leq \alpha \leq 0$ is not satisfied the theory. It is a wide range, which implies that it cannot reach a global optimum. Moreover, I think all the numerical results are strange to me, since $y$ axis is a log scale, but the variances are almost same across iterations. Please provide an answer on this in the rebuttal. ===== After the rebuttal I read the authors' rebuttal and my concerns on confidence bounds and small \alpha are resolved. I slightly increased my score.

[Author Response · NeurIPS 2020]

We thank the reviewers for the positive and constructive feedback. Below we respond to their questions.

**R1 + R4**. "**The regret curves have a very tight confidence bound, starting from the very first iterations.**
**Shouldn't it vary more? Variances are almost same across iterations?**" For the error bars (or variances), we
use the standard error: Std. Err = Std. Dev$/\sqrt{n}$, $n$ being the number of runs. In our case, $n = 15$. Division
by $\sqrt{n}$ definitely makes the error bars look smaller. We have now included an example case for Beale function
using standard deviation for error bars without diving by $\sqrt{n}$. See the plot here. We also confirm that our initial
search spaces are randomly placed across different runs and therefore, we see variances even in the beginning.
8

We would like to emphasise that, different from traditional BO al-
gorithms with fixed search space where error bars (or variances) of
regret curves tend to get tighter over time, in the context of unbounded
search space where the search space is being expanded over time,
error bars do not always have this property, they may even become
higher over time till the search spaces have not contained the global
optimum. This trend can be seen for many unbounded search space
methods such as Ha et al [8] and Vu et al [17] in our references.

**R1. "Optimization of the acq. function inside $H_t$ isn't mentioned**
**anywhere?**" We optimise acq. function for each hypercube in $H_t$ and then take the overall maximum across all
hypercubes. We did mention this in detail in our supp. material (see section "On the computational effectiveness").

**R1. "The performance of HD-HuBO against HuBO in high dimensions"**. In high dimension, HD-HuBO works
better than HuBO just because of the acquisition function optimization step *in practice*. Up to a large value of $t$, the
volume of search space $H_t$ for HD-HuBO remains much smaller than the volume of search space $X_t$ for HuBO. For
example, assuming $\alpha = -1$, $\lambda = 1$ and the dimension $d = 50$, HD-HuBO at iteration $t$ only uses $t$ small hypercubes
with size of $10\%$ of the initial search space $[0,1]^{50}$. Considering $t = 1000$, the volume of $H_t$ is at most $1000 \times 0.1^{50}$
which is at least $(1 + \sum_{j=1}^{1000} j^\alpha)^{50}/(1000 \times 0.1^{50}) \approx 5.66 \times 10^{90}$ times smaller compared to the volume of $X_t$ of
HuBO. However, despite this, HD-HuBO still attains a sub-linear convergence.

**R1 + R2 + R3. "On the comparison with GP-UCB"**. To see whether our method does better than a BO method using
a large, fixed search space, we compared our HuBO against a GP-UCB algorithm with search domain: $[-100, 100]^5$ for
the optimization of 5-dim Levy function. After 200 iters, the smallest function value found by the GP-UCB and our
HuBO were 23.10 and 2.21 respectively - a clear evidence in favor of our method. We will add these to the paper.

**R1 + R3 + R4. "Drawback of setting $\alpha = -1$; $X_0$ being far from $x^*$?**" $\alpha = -1$ and $X_0$ being far from $x^*$ make
our algorithms reach to $x^*$ slowly. However, in practice, the translation mechanism of our algorithms permits them to
jump faster toward the promising regions. We already had results shown in Figure 3 of our Sup. Material where we
used $\alpha = -1$ and set $X_0$ to be only $2\%$ of the initial search space. Our algorithms clearly outperformed all baselines.

**R2. "Why we do not start from a large domain?"**. For this approach, the crucial problem is "how large a compact
search space should be set so that $x^*$ belongs almost surely to the search domain"? Without any prior knowledge, we
should set the domain as large as possible. We consider two cases. **Case 1**: *Using a fixed search space*. In section
"Additional Results" of our Supp. Material, we already showed using GP-UCB and EI algorithms that the larger the
fixed search space, the slower is the convergence. Further, we have also compared our HuBO with GP-UCB with a
large search space $[-100, 100]^5$ and performed better. See our detail answer above in lines 27-30 in this rebuttal. **Case**
**2**: *Successively cutting down the search space*. One strategy may be to use confidence bounds UCB and LCB to cut the
search space down to a new space $S_t$ as in the algorithm branch and bound of Nando de Freitas et al (ICML 2012):
$S_t = \{x | \mu_t(x) + \sqrt{\beta_t}\sigma_t(x) > \sup \mu_t(x) - \sqrt{\beta_t}\sigma_t(x)\}$. However, $S_t$ is usually not compact and expensive to compute
in high dimensions. Further, $x^*$ only belongs to $S_t$ with probability $1 - \delta$, not probability 1, and when cutting the search
space successively, it is difficult to achieve a significant reduction from the initial large search space while maintaining
a high $1 - \delta$ across all $t$. In contrast, our algorithms do not suffer from such difficulties, easy to implement and achieve
a sub-linear rate of convergence.

**R3. "The dependence of the regret bound on $X_0$ and $l_h$"**. Theorem 4 is our main result providing the regret bound
for HD-HuBO in terms of $T$ ignoring all variables that are not the function of $T$. However, we can see the regret
bound's explicit dependence on $X_0$ (via $A$) and $l_h$ (via $\beta_t$) through Lemma 10 in Supp. material.

**R4. "On using small $\alpha$"**. We do not see a small $\alpha$ as a limitation. A small $\alpha$ is meant to slow the search space expansion
rate and in fact becomes beneficial once the search space contains the global optimum. As seen from Theorem 2, our
HuBO algorithm achieves a sub-linear regret $\mathcal{O}^*(T^{((\alpha+1)d+1)/2})$ (e.g. for SE kernel) implying that the smaller the $\alpha$,
the tighter is the regret provided $-1 \leq \alpha < -1 + 1/d$. We note that our algorithm is the only one to guarantee an exact
convergence and further with a sub-linear convergence rate despite such small $\alpha$ values.

[Meta-Review · NeurIPS 2020]

The paper has been discussed after the rebuttal that the reviewers found useful and actionable (e.g., concerns about the confidence bound). The paper is recommended for acceptance. All reviewers have acknowledged that the paper is well motivated, well written and establishes a nice interplay between theory and a practical problem of interest.